# An individualized protein-based prognostic model to stratify pediatric patients with papillary thyroid carcinoma

Zhihong Wang [1,9], He Wang[2,3,4,9], Yan Zhou[2,3,4,9], Lu Li [2,3,4,5], Mengge Lyu[2,3,4], Chunlong Wu[6], Tianen He [2,3,4], Lingling Tan[6], Yi Zhu [2,3,4], Tiannan Guo [2,3,4], Hongkun Wu [7,8,10] ✉, Hao Zhang [1,10] ✉ & Yaoting Sun [2,3,4,10] ✉

Pediatric papillary thyroid carcinomas (PPTCs) exhibit high inter-tumor heterogeneity and currently lack widely adopted recurrence risk stratification criteria. Hence, we propose a machine learning-based objective method to individually predict their recurrence risk. We retrospectively collect and evaluate the clinical factors and proteomes of 83 pediatric benign (PB), 85 pediatric malignant (PM) and 66 adult malignant (AM) nodules, and quantify 10,426 proteins by mass spectrometry. We find 243 and 121 significantly dysregulated proteins from PM vs. PB and PM vs. AM, respectively. Function and pathway analyses show the enhanced activation of the inflammatory and immune system in PM patients compared with the others. Nineteen proteins are selected to predict recurrence using a machine learning model with an accuracy of 88.24%. Our study generates a protein-based personalized prognostic prediction model that can stratify PPTC patients into high- or low-recurrence risk groups, providing a reference for clinical decision-making and individualized treatment.

Papillary thyroid carcinoma (PTC) is one of the most common endocrine malignant tumors in children and adolescents, with an incidence rate increasing by 4.4% yearly[1]. About 1.8% of thyroid cancers occur in children and adolescents, and PTC accounts for more than 90% of the cases[2]. Compared with adult PTC, pediatric PTC (PPTC) tends to have a larger tumor size, more lymph node metastases, a greater extrathyroidal extension rate, a higher distant metastasis rate, and a higher recurrence rate, while the overall mortality rate is lower. The guidelines for pediatric differentiated thyroid cancer have gaps regarding individualized diagnoses, treatments, and prognosis evaluation strategies up to the time of writing[2]. Specifically, unlike

adults, pediatric patients are not age-stratified and do not receive individualized treatments: a one-size-fits-all treatment strategy is adopted for all of them[3]. Although most PPTCs have a favorable prognosis, recurrence seriously affects patients' disease-free survival and quality of life. Because the risk factors of PPTC recurrence are not clearly identified, there is currently a lack of effective methods for evaluating the prognosis of PPTC patients and classifying them into high- or low-recurrence risk groups. Therefore, patients with a low recurrence risk may undergo aggressive surgical resections, which unnecessarily increases their risk of complications. On the other hand, patients with a high recurrence risk may receive insufficient

[1]Department of Thyroid Surgery, The First Hospital of China Medical University, Shenyang, China. [2]School of Medicine, School of Life Sciences, Westlake University, Hangzhou, China. [3]Westlake Center for Intelligent Proteomics, Westlake Laboratory of Life Sciences and Biomedicine, Hangzhou, China. [4]Research Center for Industries of the Future, Westlake University, Hangzhou, China. [5]College of Pharmaceutical Sciences, Zhejiang University, Hangzhou, China. [6]Westlake Omics (Hangzhou) Biotechnology Co., Ltd., Hangzhou, China. [7]Department of Hepatobiliary and Pancreatic Surgery, the First Affiliated Hospital, Zhejiang University School of Medicine, Hangzhou, China. [8]Zhejiang Provincial Key Laboratory of Pancreatic Disease, Hangzhou, China. [9]These authors contributed equally: Zhihong Wang, He Wang, Yan Zhou. [10]These authors jointly supervised this work: Hongkun Wu, Hao Zhang, Yaoting Sun. ✉e-mail: wuhongkun@zju.edu.cn; haozhang@cmu.edu.cn; sunyaoting@westlake.edu.cn

preoperative evaluations and postoperative monitoring, resulting in a worse prognosis.

To date, the studies on the molecular mechanism of PPTC have been mostly limited to the genetic level[4–6]. They mainly focused on analyzing the PPTC etiology and providing a benign versus malignant diagnosis but did not produce tools for a personalized prognosis evaluation. Compared with adult PTC, PPTC is characterized by a higher prevalence of gene rearrangements and a lower frequency of point mutations in the proto-oncogenes implicated in PTC. Specifically, *BRAF* mutations are rarer, while *RET/PTC* rearrangements and gene fusions are more common in pediatric than in adult PTC[4,7]. Consequently, these differences may affect the efficacy of gene-based diagnosis and prognosis evaluations of pediatric thyroid cancer.

Compared to genes, proteins could provide a more valuable contribution to the prognosis evaluation of diseases because they are the final products of gene expression[8]. However, the proteomic changes caused by PPTC remain unknown. Our previous study showed the potential of a machine learning-assisted proteomic analysis to discriminate between benign and malignant thyroid nodules[9,10]. Additionally, as we used trace samples from formalin-fixed paraffin-embedded (FFPE) tissues[11], we showed the feasibility of such a study using preoperative fine-needle aspiration (FNA) samples which contain only thousands of cells and are hard to be analyzed by ordinary proteomic technology[9].

In this work, we profile the proteomic characteristics of PPTCs and compare them with pediatric benign nodules and adult PTCs. The immunity-related pathways and functions are significantly altered in PPTC, as indicated by dysregulated protein analysis. Moreover, nineteen of the dysregulated proteins have been selected by a customized model to stratify pediatric patients into high- or low-recurrence risk groups, which achieves an accuracy of 88.24%. Our study provides a way to stratify pediatric patients with different recurrence risks, which may be a reference for clinical decision-making and individualized treatment.

## Results

### Clinical characteristics of our study population

The overall study design is demonstrated in Fig. 1a. We enrolled 85 PPTC patients (PM) and 83 pediatric patients with benign nodules (PB) (Fig. 1b), and their clinicopathological features were summarized in Supplementary Data 1. This cohort included 23 males and 62 females (male-to-female ratio of 1:2.7) with an average age of $15.6 \pm 2.4$ years and 15 males and 68 females (male-to-female ratio of 1:4.5) with an average age of $15.9 \pm 1.9$ years in PM and PB groups, respectively. All patients were admitted to the hospital with a mass in the neck, and their average tumor size was $2.4 \pm 1.3$ cm in PM group, which is smaller than those in PB group $3.8 \pm 1.3$ cm. In PM group, the median follow-up time was 71 months (interquartile range 48–113), during which no death was reported. Lung metastasis was discovered in one patient before the operation, and no change was reported after the radioactive iodine (RAI) therapy. Postoperative structural recurrence occurred in 12 cases (average age of 14): ten ipsilateral cervical lymph node metastases and one contralateral cervical lymph node metastase. One case developed postoperative lung metastases. All the cases of lymph node metastases were reoperated. During the follow-up evaluations, we found that the lesions of the patients with lung metastases had shrunk after the RAI therapy, and no growth or mental retardation was detected in any patient. Finally, no hematological or other secondary solid primary tumors were found during the postoperative follow-up.

### Three clinical features are the risk factors for PPTC recurrence

To identify the clinical recurrence risk factors in our study cohort, we built a univariate Cox proportional hazard (CoxPH) model for each of the eleven clinical features collected for the PM patients ($N = 85$). Age ($P = 0.0174$, hazard ratio (HR) = 0.7928, 95% confidence interval (CI): 0.6547–0.96), total lymph node metastasis number (TLNN, $P = 0.0225$, HR = 1.076, 95% CI: 1.01–1.146), and lateral lymph node metastasis number (LLNN, $P = 0.0111$, HR = 1.101, 95% CI: 1.022–1.185) had $P$ values smaller than 0.05. Next, we split the PM patients into two groups (< or ≥ the median) based on each significant factor. Further analyses showed significant differences between the Kaplan–Meier survival curves of the two groups for these three features (Fig. 2a). Moreover, when treated as a categorical variable (0 representing ages below the median (16-year-old), 1 otherwise), age was more significantly associated with recurrence ($P = 0.0302$, HR = 0.2645, 95% CI: 0.0794–0.8804). Our results showed that age, TLNN, and LLNN may be risk factors for recurrence in pediatric patients.

To determine the form of the age variable, the eleven clinical features were next used as the inputs of multivariate CoxPH models. In particular, we input age as either a continuous integer or a categorical variable (as described in the previous paragraph). Our forest plot showed the HRs for eleven clinical features, indicating the positive or negative influence of each feature on the PPTC recurrence. Only categorical age was almost related to PPTC recurrence. The global $P$ value (log-rank), the Akaike information criterion (AIC), and the Concordance Index (C-Index) outperformed when age was used as a categorical variable (Fig. 2b, c). Therefore, age was determined as a categorical variable for the downstream analyses.

### More than 10,000 protein qualifications with high quality

We collected 234 thyroid tissues from three groups: 85 PM, 83 PB, and 66 adult malignant (AM) nodules. We randomly selected two samples and conducted one more replicate in each group. The resulting 240 tissues were randomized into 16 batches with 15 tissue samples each, and one pooled sample was used as a linker for the batches. Among them, we quantified 10,426 proteins (Supplementary Data 2).

To reduce the statistical bias, we removed 1272 (-12.2%) proteins with a missing value (NA) rate above 85%. This resulted in a final dataset including 9154 proteins. Quality control analysis showed the coefficient of variations (CVs) of the proteins across the 16 pooled samples were mainly between 0.0 and 0.2, with a median of 0.0493 (Supplementary Fig. 1a); the CVs of the proteins across each pair of replicates were mostly less than 0.2, with medians of 0.0662, 0.0947, 0.1238, 0.0890, 0.0645 and 0.1123 (Supplementary Fig. 1b). These results indicate the high stability of our instrument and the reliability of our data (Supplementary Fig. 1c, d).

To reduce the influence of NAs and batch effects, we imputed the NAs and then used ComBat to adjust for batch effects, and, by visualizing the resulting data using two-dimensional Uniform Manifold Approximation and Projection (UMAP), no noticeable batch effects could be detected (Supplementary Fig. 1e). After these preprocessing steps, there were 240 samples (87 PM, 85 PB and 68 AM) and 9154 proteins left (Supplementary Fig. 1f).

### Proteomic differences among pediatric malignant, pediatric benign and adult malignant thyroid nodules

To further explore the differences between PM and PB/AM, we determined the dysregulated proteins and generated two volcano plots showing 243 (PM vs. PB) and 121 (PM vs. AM) differentially expressed proteins (DEPs) with fold change (FC) > 1.5 and adjusted $P < 0.05$ (Fig. 3a, b and Supplementary Data 3). The DEPs with FC > 1.5 from the two pairwise comparisons were distributed in the scatter plot, and 27 proteins were co-up/downregulated in PM vs. AM and PM vs. PB (Fig. 3c). Furthermore, the expression of 37 selected proteins shown in the heatmap, which was from the co-dysregulated proteins and the top-five up- and downregulated proteins in the two pair-wise comparisons (Fig. 3d). According to the enrichment analysis of annotated keywords performed using STRING database, the most upregulated proteins in PM, compared to the other two groups, were involved in MHC-II and immunity (Supplementary Table 1). These results show that

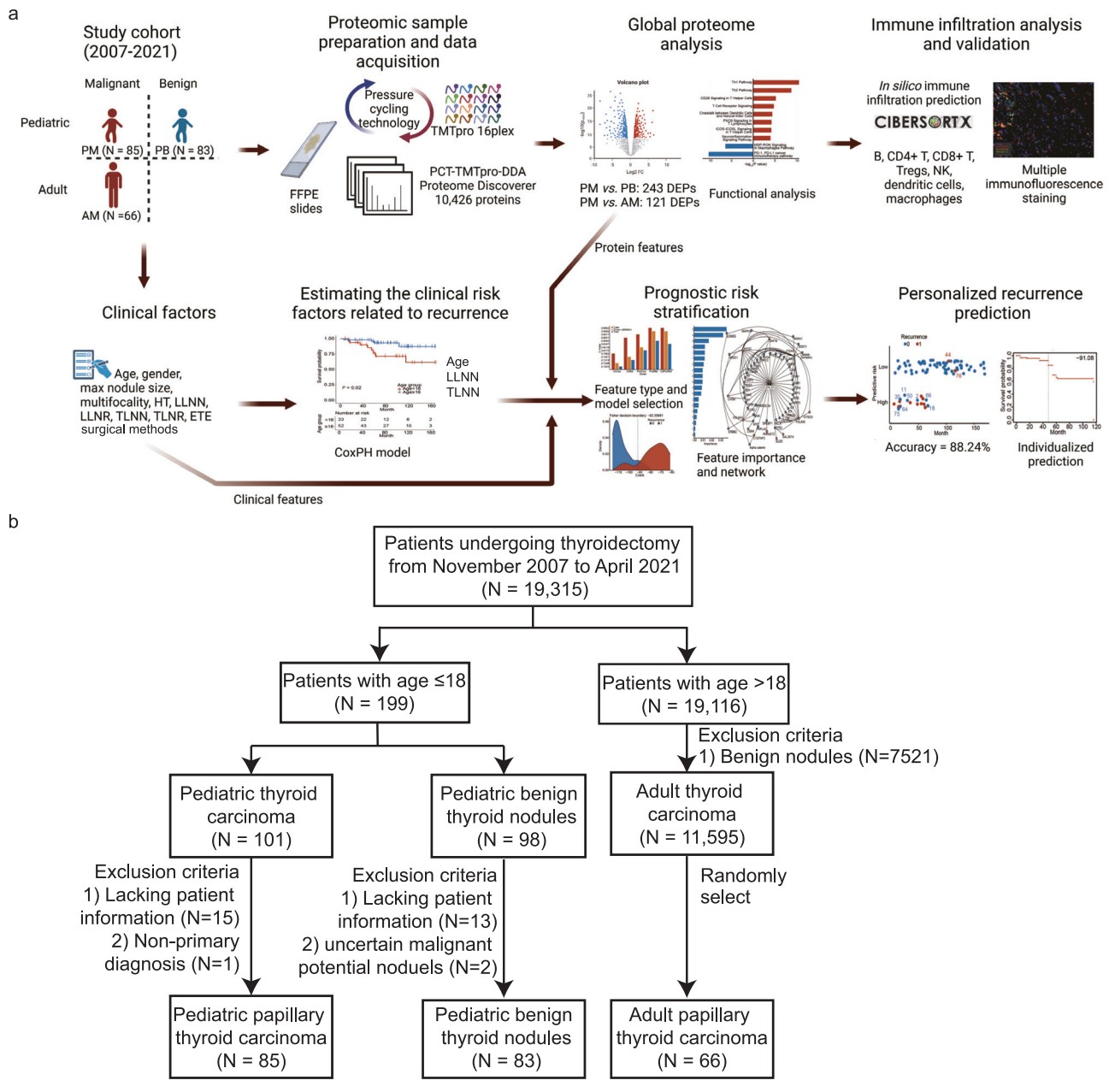

**Fig. 1 | Study overview. a** Study design of analyzed cohort and experiment workflow. Created through Biorender.com. **b** Enrollment and exclusion criteria for pediatric papillary thyroid carcinoma (PTC), pediatric benign nodule and adult PTC patients.

PPTC has a unique protein expression that differs from pediatric benign nodules and adult PTC.

Next, the functions and pathways enriched for DEPs with 1.5 FC in the PM and the PB groups were almost all related to immune system regulation: mainly functions pertaining of T cells and natural killer (NK) cells (Fig. 3e, f and Supplementary Data 4). Then, the comparison of the PM with the AM group further showed pediatric thyroid cancer was associated with the regulation of inflammatory or immune-related pathways (Fig. 3g). These results suggest the development of PPTC is related to altered immune system functions.

**Immune infiltration and expression level of immune checkpoints in pediatric thyroid nodules**

Since multiple dysregulated immune-related pathways and biological processes were enriched, we further conducted an analysis of immune infiltration in the pediatric samples using 'in silico flow cytometry' CIBERSORTx[12]. Seven types of immune cells were imputed, and their relative proportions are shown in Fig. 4a. The fractions of CD8+ T cells ($P = 3.7 \times 10^{-12}$), macrophages ($P = 0.031$), dendritic cells ($P = 1.4 \times 10^{-5}$) and Treg cells ($P = 0.007$) vary significantly. CD8+ T cells and macrophages increased in PM samples, while dendritic cells and Treg cells decreased in PM samples. To validate immune infiltration results from in silico analysis, we processed immunofluorescent staining for CD4+ and CD8+ T cells, which marked CD3+/CD4+ and CD3+/CD8+, respectively. Representative staining images of enriched CD8+ T cells and decreasing CD4+ T cells in the PMs are shown in Fig. 4b.

To further explore the tumor immune microenvironment, we compared the abundances of immune checkpoint proteins between PB vs. no-recurrence PM (PM-NR) and PM-NR vs. recurrence PM (PM-R). Among the 31 immune checkpoints quantified in our proteome data, poliovirus receptor (PVR) and interleukin 10 receptor B (IL10RB) had significantly lower levels in the most aggressive group PM-R (Fig. 4c). No immune checkpoint proteins were found upregulated with an increasing malignancy within PB, PM-NR and PM-R groups.

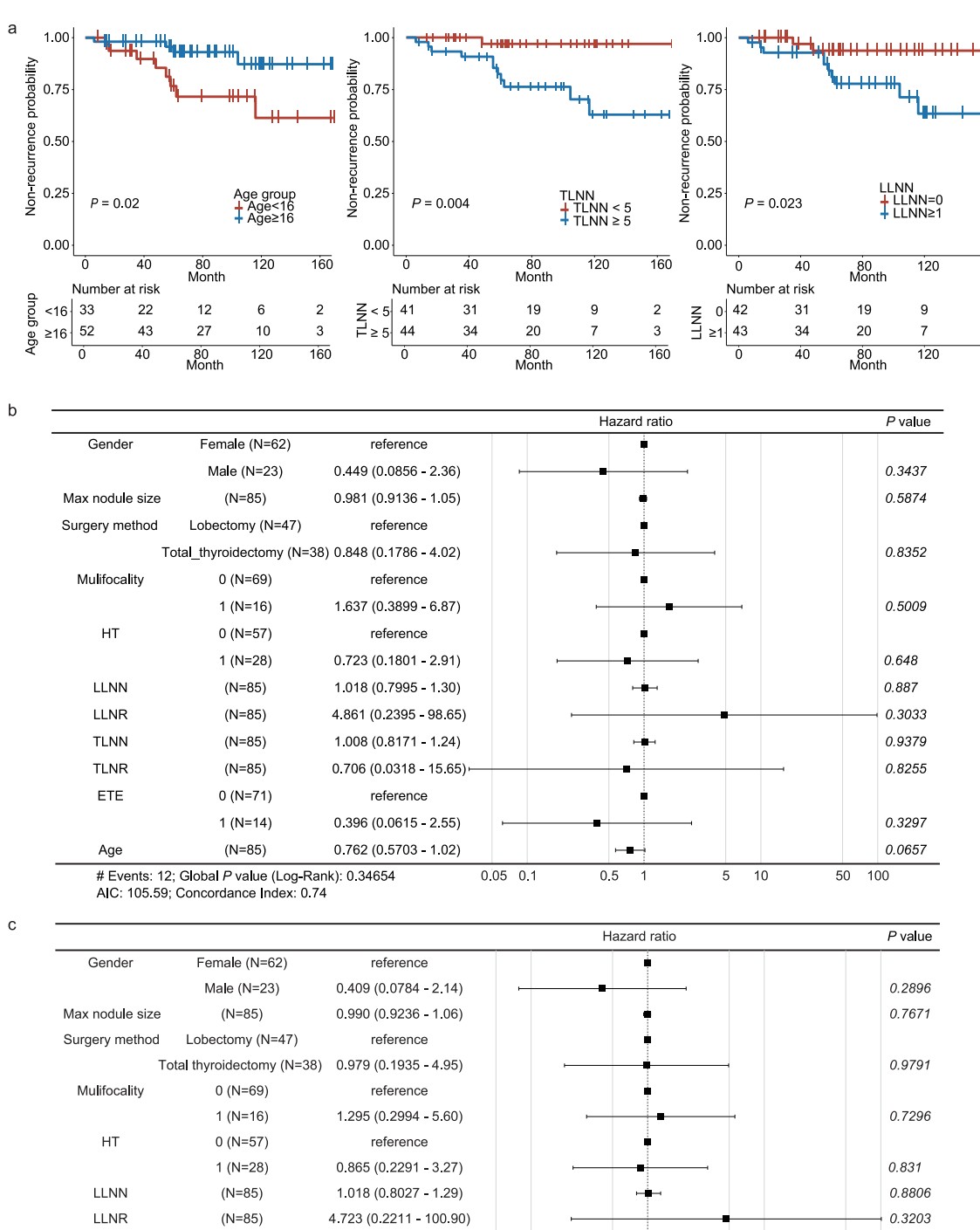

**Fig. 2 | Analysis of the clinical recurrence risk factors for pediatric papillary thyroid carcinoma (PPTC). a** Kaplan–Meier survival curves of two groups (red: patients below the median; blue: otherwise) for three significant factors: age, total lymph node metastasis (TLNN) and lateral lymph node metastasis number (LLNN). *P* values are derived from Log-Rank Test. **b**, **c** Forest plots for two multivariate CoxPH models using (**b**) continuous non-negative integer age and (**c**) categorical age, respectively. Data are presented as hazard ratio value with 95% confidence interval. *P* values are tested by Cox proportional hazard model.

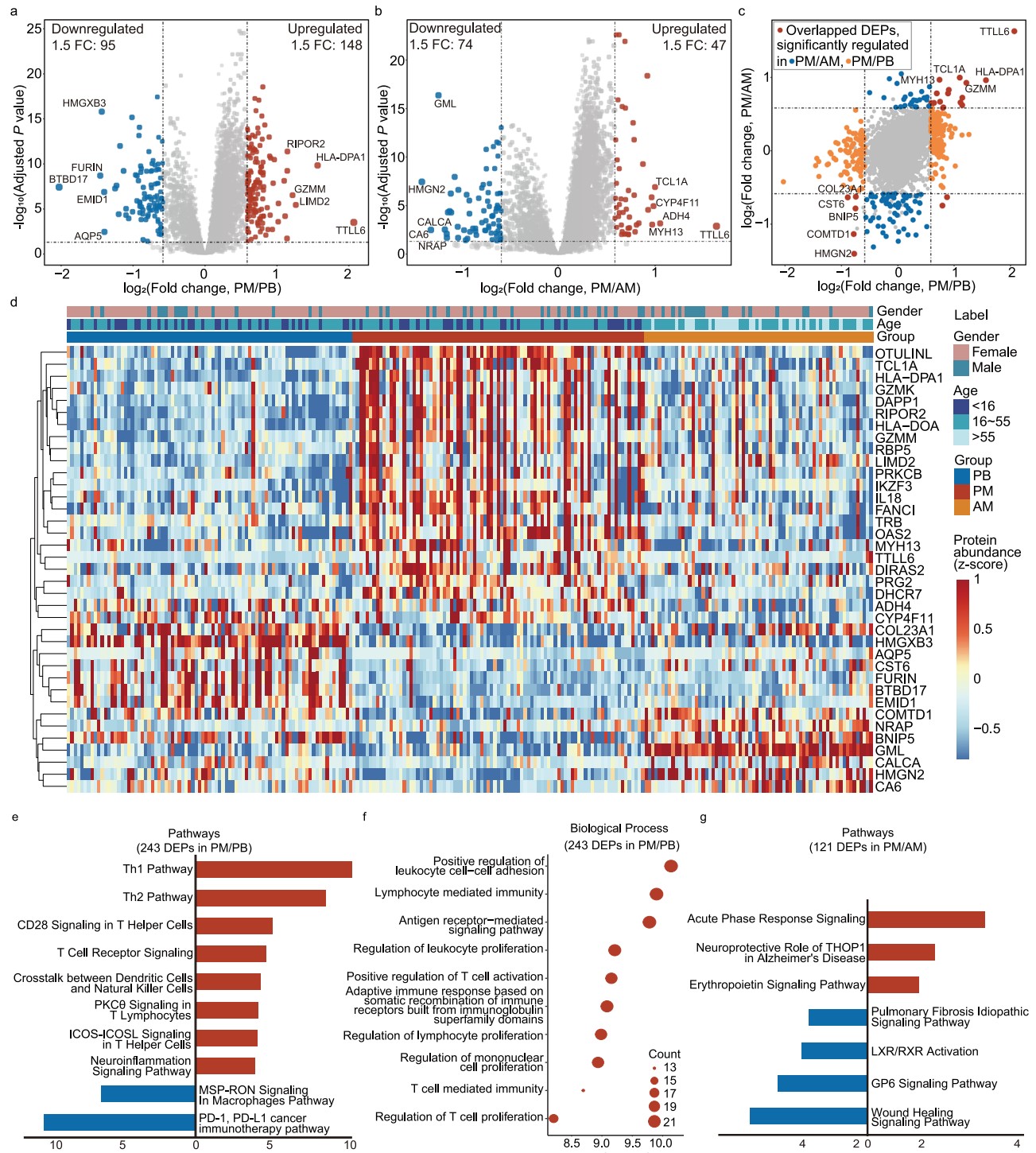

**Fig. 3 | Functional analyses of the dysregulated proteins. a**, **b** Differentially expressed proteins (DEPs) are shown in the volcano plots: **a** pediatric malignant nodules (PM) vs. pediatric benign nodules (PB) and **b** PM vs. adult malignant nodules (AM). The cutoff is defined by requiring the fold change (FC) to be greater than 1.5, with adjusted $P < 0.05$ (BH-adjusted two-sided Welch's $t$ test). The names of the up/downregulated proteins with the top five largest FC are reported in the plots. **c** The scatter diagram shows the FC distribution of the dysregulated proteins in two pairwise comparisons: PM vs. PB and PM vs. AM. The overlapping significantly co-dysregulated proteins are colored in red. The proteins significantly dysregulated in PM/PB are colored in orange, and those dysregulated in PM/AM are colored in blue. Here, the DEP lists were derived from FC threshold 1.5. **d** The

heatmap shows 37 proteins: they are co-upregulated/co-downregulated proteins and the five most up-/down- regulated ones from the volcano plots (**a**, **b**). Proteins were clustered using hierarchical clustering. **e** Pathway enrichment of the 243 DEPs from the volcano plot with the PM/PB comparisons (FC > 1.5). The red and blue bars represent the active and inhibited pathways, respectively. **f** Results of the gene ontology enrichment of the biological processes using the DEPs in PM/PB. **g** Pathway enrichment of 121 DEPs from the volcano plot with the PM/AM comparisons (FC > 1.5). The red and blue bars represent the active and inhibited pathways, respectively. $P$ values are derived from one-sided Fisher's Exact Test for pathway and gene ontology enrichment.

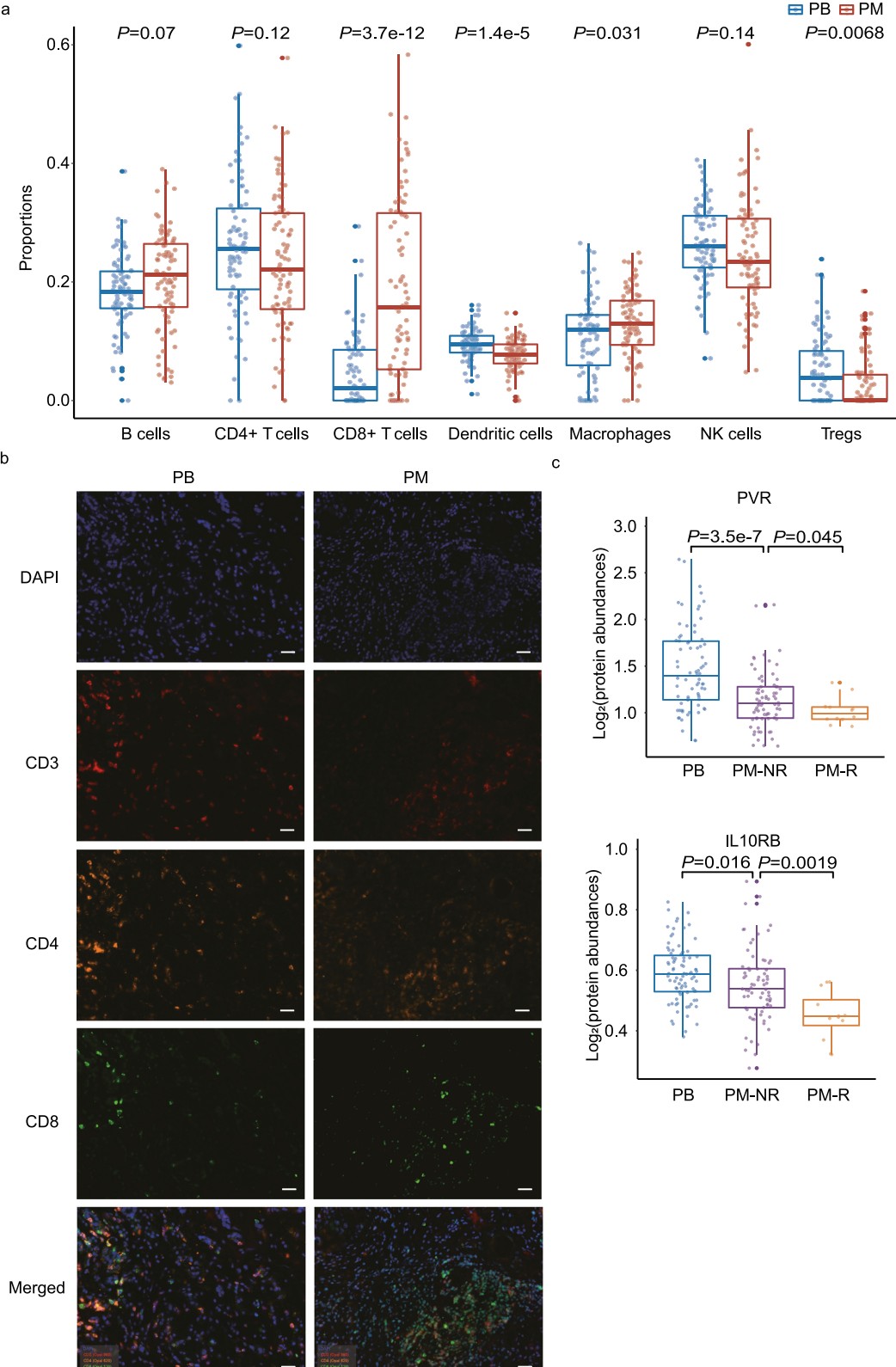

**Fig. 4 | In silico immune infiltration analysis and expression levels of immune checkpoints. a** Relative proportions of seven types of immune cells in pediatric benign (PB, $N = 83$) and pediatric malignant (PM, $N = 85$) samples imputed by CIBERSORTx. Boxes are first and third quartiles, the center line is median, whiskers are ±1.5 interquartile range, and dots are individual data points. Abundance outliers and missing values are not included in the boxplot, throughout Fig. 4.

**b** Representative multiplex immunohistochemistry staining in PB ($N = 10$) and PM samples ($N = 10$). The scale bar represents 50 μm. **c** The protein expression abundances of poliovirus receptor (PVR) and interleukin 10 receptor B (IL10RB) in PB ($N = 83$), PM-NR (non-recurrence, $N = 73$) and PM-R (recurrence, $N = 12$) groups. The significance throughout Fig. 4 is determined by two-sided Welch's $t$ test.

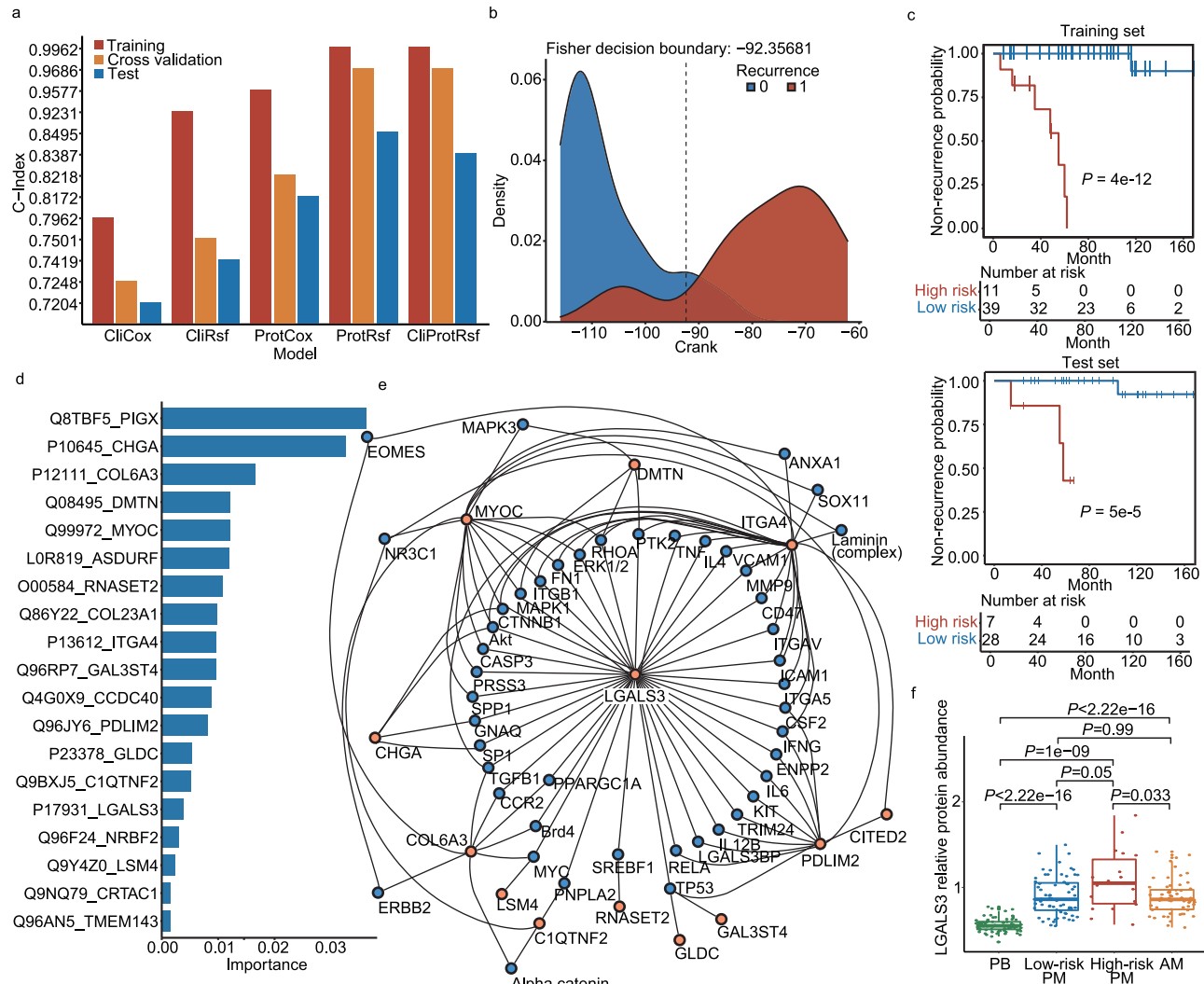

**Fig. 5 | Pediatric papillary thyroid carcinoma (PPTC) prognostic prediction.**
**a** The C-indexes of our five models were calculated on training, threefold cross-validation, and test sets. **b** Density curves of the training continuous risk ranking (Crank) scores of two groups (recurrence (1) or no recurrence (0)). The Fisher decision boundary was used to differentiate the low- from the high-risk groups. **c** The Kaplan–Meier survival curves of the low- and high-risk groups, calculated on the training and test sets, show significant differences. **d** Permutation importance of the 19 proteins from the ProtRsf model. **e** Network showing the 19 features of the ProtRsf model with the connected proteins enriched using the Ingenuity Pathway Analysis software. **f** The relative protein abundances of galectin-3 (LGALS3), the hub protein of the network (**e**), in the four groups. Boxes are first and third quartiles, the center line is median, whiskers are ±1.5 interquartile range, and dots are individual data points. Abundance outliers and missing values are not shown in the boxplot. Biologically independent samples shown in boxplot: PB, $N = 77$; Low-risk PM, $N = 67$; High-risk PM, $N = 18$; AM, $N = 60$. The mild outliers were removed, and a two-sided unpaired Wilcoxon rank-sum test was used, without continuity correction, to calculate the $P$ values.

## Development of PPTC prognostic prediction models and individualized prognostic stratification

To predict the PTC recurrence risk of patients from the PM group, the PM samples were randomly divided into a training set ($n = 50$, ~60%) and an independent test set ($n = 35$, ~40%). Then, we developed five models based on two algorithms (Cox proportional hazard model and random survival forest) and two types of features (clinical features and proteins). Specifically, we developed the following models: two Cox proportional hazard models based on clinical features (CliCox) or protein features (ProtCox); three random survival forests based on clinical features (CliRsf), protein features (ProtRsf), or clinical and protein features (CliProtRsf). For each model, we tuned the hyperparameters using grid search strategy and three-fold cross-validation, selected the features, and trained the model using the training set. The final hyperparameter settings of the five models are summarized in Supplementary Table 2. The ProtRsf model was the best-performing one as it achieved the highest C-index values: 99.62%, 96.86%, and

84.95% on the training, the cross-validation, and the independent test sets, respectively (Fig. 5a). Notably, the combination of features used by CliProtRsf only contained 21 proteins without any clinical features, which means the clinical features did not contribute to the model's prediction significantly when protein features existed. The clinical features even interfered with the protein features; thus, more proteins were needed to compensate for this effect. However, even with more protein features selected for the model, CliProtRsf did not outperform ProtRsf (containing 19 proteins) in C-index. Therefore, we chose the ProtRsf model for our downstream analyses.

Then for each patient, we predicted his/her individualized survival curve firstly and deduced the Crank risk score. Then, as shown in Fig. 5b, using the training set, we determined the risk stratification threshold according to the risk scores of the recurrent and the non-recurrent patients. Therefore, the PM patients were classified as high or low-risk according to this threshold. The Kaplan–Meier curves of the high- and low-risk patients differed significantly in the training and

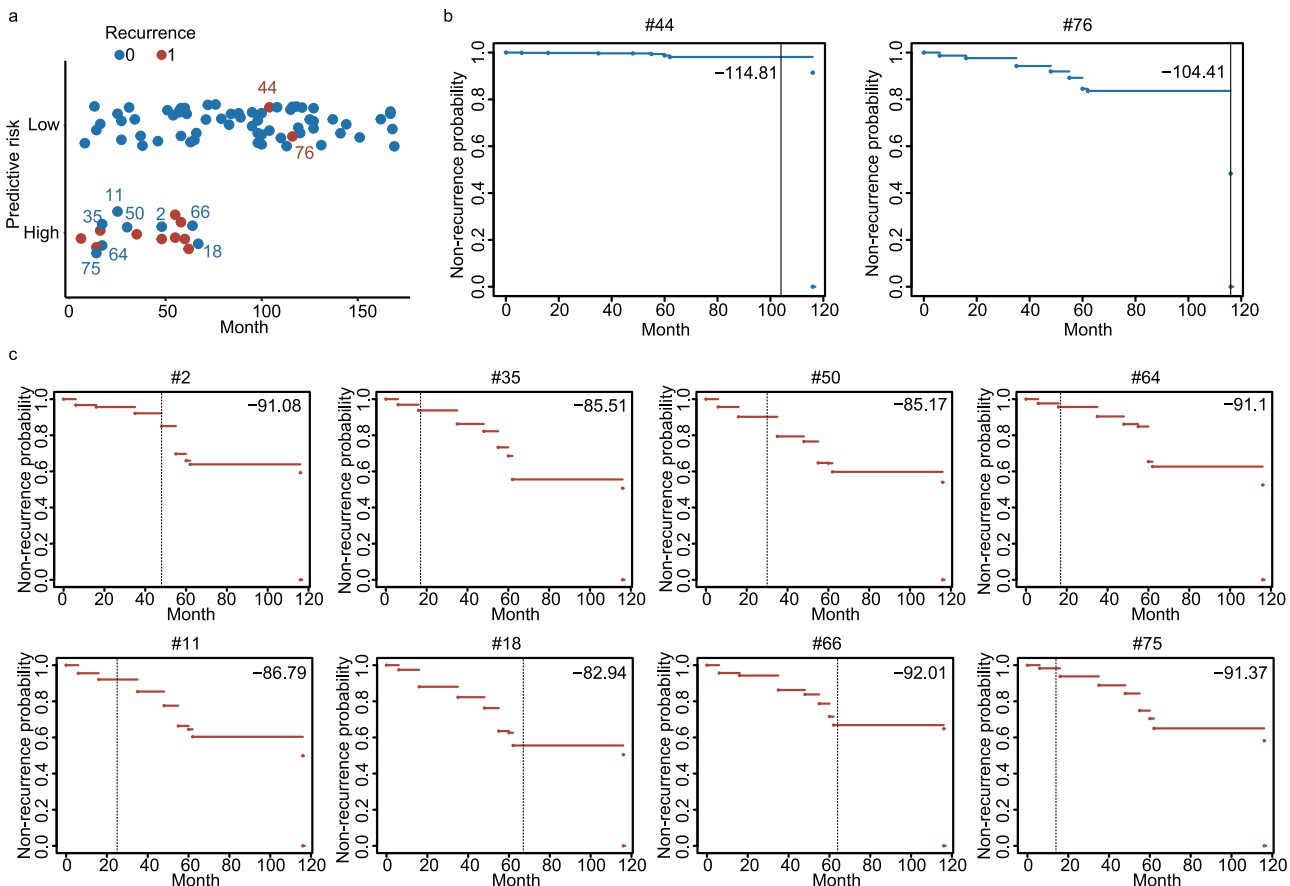

**Fig. 6 | Risk stratification. a** Predicted risk stratification for pediatric papillary thyroid carcinoma (PPTC) patients. The sample indexes of false positives ($N = 8$) and the false negatives ($N = 2$) are labeled. **b, c** The predicted survival curves of the two false negatives (**b**) and eight false positives (**c**) with their continuous risk ranking (Crank) scores, sample indexes and recurrence or latest follow-up times (shown by the vertical lines).

independent test sets, indicating the strong generalization capability of our model (Fig. 5c).

**Analysis of 19 feature proteins**

The random survival forest algorithm selected 19 proteins as features for the ProtRsf model; the importance of these proteins is shown in Fig. 5d. Of these 19 proteins, five have already been reported in thyroid cancer studies, including galectin-3 (LGALS3)[13], chromogranin-A (CHGA)[14], collagen alpha-3(VI) chain (COL6A3)[15], collagen alpha-1(XXIII) chain (COL23A1)[16], and integrin alpha-4 (ITGA4)[17]. Furthermore, myocilin (MYOC) has been linked to the thyroid's function[18] (Supplementary Table 3). The remaining 13 proteins have not yet been reported associated with thyroid disease.

Our network analysis showed that 13 of the 19 protein features were directly or indirectly connected. In particular, LGALS3, the hub protein, may perform a significant role in pediatric thyroid carcinoma (Fig. 5e). The protein abundance of LGALS3 in four groups (PB, PM low-risk, PM high-risk, and AM) is shown in Fig. 5f with Wilcoxon $P$ values. LGALS3 has the lowest expression in the PB group, with significant differences compared to the expression in the other groups. In contrast, its expression was the highest in the PM high-risk group. These results show that a high LGALS3 expression may be associated with a higher recurrence risk. Moreover, we conducted transcription regulator prediction and found four transcription regulators enriched with $P < 0.01$ (Supplementary Table 4). From them, sterol regulatory element binding transcription factor 1 (SREBF1) is a reported, prognostically relevant protein in thyroid cancer[19,20].

To explore if the 19 proteins selected by the ProtRsf model were related to the immune system, we calculated Pearson correlations of immune cell fractions and the 19 proteins in PM high- and low-risk groups, respectively (Supplementary Fig. 2). ITGA4 ($P = 7.28 \times 10^{-4}$ and $P = 2.05 \times 10^{-8}$ in high- and low-risk groups, respectively) and GAL3ST4 ($P = 1.56 \times 10^{-4}$ and $P = 1.24 \times 10^{-4}$ in high- and low-risk groups, respectively) were found positively correlated to CD8+ T cells in both groups. For the 31 immune checkpoint proteins quantified, only the abundance of IL10RB was found to decrease with the predicted recurrence risk and highest in PB samples ($P = 0.0012$ and $0.038$, respectively; Supplementary Fig. 3).

**Overall and individualized performance of the 19-protein model**

We next evaluated the efficacy of our ProtRsf model in stratifying PPTC patients into groups with a high or low risk of recurrence. The model could correctly predict the prognosis of 75 cases of our 85 PM patients with an accuracy of 88.24% (Fig. 6a). However, ten patients were wrongly classified: two were false negatives and eight were false positives. The predicted prognostic survival curves of each misclassified patient are shown in Fig. 6b, c.

Then, we carefully analyzed the ten wrong predictions. The two false negative events corresponded to patients who underwent a recurrence but were classified as the low-risk group by the model. However, their recurrences were detected after 104 and 116 months, which were much longer than the median follow-up time (71 months) (Fig. 6b). For the false-positive patients, the follow-up times (14, 17, 17, 25, 30, 48, 64, and 67 months) were all shorter than the median follow-up time (71 months) (Fig. 6c). These patients were only follow-up for a

short period when we started this study, which means they may go through recurrence in the future.

## Discussion

PPTC is the most common endocrine malignant tumor in pediatric patients, which exhibits different clinical characteristics from adult PTC. There is still no effective strategy for evaluating the recurrence risk of pediatric thyroid carcinoma. In our study, we collected 85 PM, 83 PB, and 66 AM thyroid nodule tissues from 234 patients. Using labeled quantitative proteomic technology, we measured 10,426 proteins, to our knowledge, which is a large-scale proteomic study on pediatric thyroid cancer patients. It is also valuable data with considerably deeper quantifications (more than 10,000 proteins) in thyroid nodules compared with previous studies which detected thousands of proteins[9,21,22]. We next found that immune processes were upregulated in PM nodules. Finally, we generated a model capable of predicting the recurrence risk of PM patients.

From our clinical data, we found age and lymph node metastases were important prognostic indicators of PPTC which are matched with previous findings[23–26]. In our PM group, the age of 16 was the cutoff for predicting recurrence-free survival (RFS), as nine of the 12 recurrent patients were younger than 16. Furthermore, 69 cases (81.18%) from our PM group had total lymph node metastases. Additionally, we found that TLNN and LLNN correlate with RFS. However, unlike previous studies[24], the lymph node metastasis rate of our study cohort did not suggest recurrence, which may be due to the different number of lymph node dissections. Although several factors were shown to be related to poor prognosis, however, we found that the risk factors derived from clinical indicators are only suggestive of clinical phenomena and are, thus, insufficient for formulating prognostic predictions and risk stratification by the model.

It has become a trend for molecular detection to apply to tumor risk stratification according to the latest version of the World Health Organization published in 2022[27]. Many studies have suggested that gene expression and clinical features of PTC were different between children and adults which is related to different prognosis[4,28]. Whereas gene correlation analyses can explain, to a certain extent, the difference in clinicopathological features between pediatric and adult thyroid cancers, current genomics studies have a limited role in the risk stratification of PPTC. Therefore, we chose to use proteomics data as the base of our predictions since proteins are the biological activity effectors. Using proteomic data, our predicting model achieves higher performance in predicting recurrence risk. Even when we combined proteins with clinical features as candidates, the model did not select any clinical features. The panel of proteins evolved by the model is significantly more accurate for predicting PPTC prognoses than clinical features.

Among the 19 proteins, CHGA[14], COL6A3[15], COL23A1[16,29], ITGA4[17] and LGALS3[13] have been reported to be associated with thyroid cancer, and MYOC[18] is related to thyroid function. Notably, LGALS3 is an important marker located in the core of the network (Fig. 5e). Its inhibitor inhibits apoptosis resistance and the invasion of thyroid cancer cells through the AKT/β-catenin pathway[30]. In agreement with these previous findings, in our study, the expression of LGALS3 in the PM high-risk group was significantly higher than in the PM low-risk group and the AM group. The high expression of LGALS3 might promote cancer invasion and impede the function of the immune system to make the cell apoptosis, leading to cancer recurrence.

Based on the 19-protein panel, our ProtRsf model achieved an accuracy of 88.24% in stratifying PPTC patients into groups with a high or low risk of recurrence (Fig. 6a). Although the high performance we got, ten patients were wrongly classified. We next have carefully investigated the mispredicted samples (Fig. 6b, c). The eight false-positive samples are patients who are predicted to relapse but have not yet done so. These patients have a relatively short current follow-up period and, in terms of survival curves, each of them has a low risk of recurrence as of the current follow-up time, but their probability of recurrence at five years after surgery will increase substantially as time continues to progress, as shown in the predicted prognostic survival curves (Fig. 6c). It is therefore difficult to be sure that the model is predicting them incorrectly at this time and close follow-up is still needed for these patients to allow time to give us the true answer. For the two false negatives, the recurrence intervals are both more than 100 months which is much longer than the median follow-up time (71 months), to some extent, it also represents relatively inert biological behavior.

The tumor immune microenvironment also plays a key role in the development and progression of thyroid cancer[31]. Most studies of the immune microenvironment of PTC have focused on adults rather than children and adolescents. In our study, we showed the 243 DEPs between PM and PB patients are closely related to immune dysregulation. Additionally, a high level of PD-L1 is associated with poor prognoses, such as an increased risk of thyroid cancer recurrence and lymph node metastases[32–36]. The results imply that dysregulated immune cell compositions and altered immune monitoring may play crucial roles in PTC genesis in pediatric patients.

CD8+ T cells recognize tumor cells which express tumor antigens and attack by inducing cell death[37]. In adult PTC, CD8+ T cells were found to have a higher frequency than in benign samples[38], and the infiltration of CD8+ T cells was related to increasing disease-free survival[39]. Our data showed higher levels of CD8+ T cells infiltration in PM than in PB, which is consistent with adult patients. CD4+ T cells were not found to be significantly different between PB and PM. Similarly, the functions of these cells in tumor prognosis were not found[39,40].

The findings of this study have to be seen in the light of some limitations. This is a retrospective study in a single center; therefore, future studies will validate the model on preoperative prospective samples in more centers to cover the diversity of the samples. Also, our results need to be validated with a larger cohort and longer follow-up time to evaluate our model's generalization. Despite these limitations, we have shown the feasibility and importance of using proteomics data for the stratification and prognostic prediction of PPTC patients.

Proteomics offers, among others, the advantages of high-throughput quantification and microsampling, the latter enabling clinical applications with preoperative FNA samples. With this method, we can make high- and low-risk stratification assessments and distant metastasis predictions before the operation, guide the resection scope during the operation, evaluate the prognosis after the operation, and formulate individualized follow-up strategies. Additionally, integrating multidimensional data (i.e., ultrasound images, gene information, blood tests, etc.) can depict the state of the tumor more comprehensively and view the tumor from different perspectives, thus obtaining a more accurate assessment, which of course cannot be achieved without the support of big data and artificial intelligence.

In conclusion, we generated a protein-based personalized prognostic prediction model that could stratify pediatric patients with PTC, providing a reference for clinical decision-making and individualized treatment.

## Methods
### Study population
This study protocol and waiver of informed consent were approved by the Ethics Committee of the First Hospital of China Medical University with the study number 2021-287-2. In this retrospective study, we evaluated pediatric patients (≤18 years) with thyroid nodules, including 85 PM and 83 PB thyroid nodules, who underwent surgery in the First Hospital of China Medical University between November 2007 and April 2021.

The exclusion criteria for PM were the following: (a) with a history of radiation exposure or family history, (b) with poorly differentiated PTC, (c) loss of follow-up or incomplete clinical data, and (d) non-primary operation. We excluded uncertain malignant potential nodules for the PB group. We also included 66 AM patients with PTC to compare pediatric and adult thyroid cancer proteomic profiling. The detailed pediatric patient characteristics are listed in Supplementary Data 1.

Preoperative pulmonary computed tomography (CT) showed that one patient had multiple metastases in the lung. All patients were surgically treated. Lobectomy and ipsilateral central lymph node dissections were performed in unilateral PTC. Total thyroidectomy was performed in patients with ETE, such as the invasion of nerves, blood vessels, or trachea. Patients with bilateral PTC underwent total thyroidectomy and bilateral central lymph node dissections. For PM patient group, 47 (55.29%) underwent lobectomy, and 38 (44.71%) had a total thyroidectomy. We recorded 16 cases (18.82%) with multifocal disease, 69 (81.18%) with lymph node metastases and 43 cases (43/85, 50.59%) of lateral cervical lymph node metastases in PM group. Postoperative treatment included thyroid-stimulating hormone inhibition and RAI therapies.

After the surgery, the patients were required to have follow-up visits every 3–6 months for the first year through cervical ultrasounds and thyroid functional examinations. The re-examination interval was then prolonged for patients with negative ultrasounds or CT, low serum thyroglobulin level, or no persistent disease. Disease remission and recurrence were determined according to the American Thyroid Association management guidelines[2,41]. Disease remission was defined as two consecutive negative whole-body scans and ultrasounds. Due to inaccurate evaluation based on serum Tg and TgAb for patients with lobectomy, only structural recurrence was considered in this study. Disease recurrence was determined as a new disease in the thyroid bed or lymph nodes proven by cytology or histopathology, and/or confirmed by ultrasounds or CT scans, or distant metastases detected by whole-body scan.

## Experimental design and statistical rationale

The overall study design was illustrated in Fig. 1a. We collected FFPE slides for proteomics data acquisition. Each slide was stained with hematoxylin and eosin and reviewed by at least two experienced histopathologists and histopathological subtypes for PM were further evaluated according to The 15th edition of the World Health Organization Classification of Endocrine and Neuroendocrine Tumors[42]. From the 85 patients, there are 69 cPTC (81.2%), eight diffuse sclerosing variant PTC (9.4%), six hobnail variant PTC (7.1%), one solid variant PTC (1.1%) and one columnar cell variant PTC (1.1%). For the twelve patients with recurrence, ten of them are cPTC, one diffuse sclerosing variant PTC and one hobnail variant PTC. Each slide was reviewed and processed to make sure the tumor ratio was approximately more than 80% before proceeding to proteomic sample preparation.

We collected 240 thyroid nodules FFPE slides (10 μm thickness) from 234 patients (85 PM, 83 PB, 66 AM). Two samples from each group were randomly selected as replicates. To minimize the potential artificial effects during experiments, we randomly allocated the 240 tissues into 16 batches. In each batch, there were 15 tissue samples and one pooled sample was used as an internal reference scaling for the batches. The replicates and pooled samples were analyzed for data quality control.

## Recurrence risk factors among clinical features

To identify recurrence risk factors among the clinical features of the PTC pediatric patients, we conducted univariate and multivariate analyses using eleven clinical features. In particular, we used the Cox proportional hazard (CoxPH) model and combined prognosis information: recurrence events, the time interval between surgery and recurrence, or between surgery and the last follow-up. The eleven clinical characteristics were: age, gender, maximum nodule size, multifocality, ETE, total lymph node metastasis rate (TLNR), lateral lymph node metastasis rate (LLNR), total lymph node metastasis number (TLNN), lateral lymph node metastasis number (LLNN), surgical methods, and Hashimoto thyroiditis (HT).

We built a univariate CoxPH model for each clinical feature, identified the factors whose P values were less than 0.05, split the PM patients into two groups (< or ≥ the median value) based on each significant factor, and compared the Kaplan–Meier survival curves of the two groups. Next, we transformed the age factor from a continuous non-negative integer to a categorical variable (0 representing ages below the median value (16-year-old), one otherwise) and performed the same analysis. Lastly, the eleven clinical features were input into the multivariate CoxPH model two times: using the continuous non-negative integer age or the categorical age. We then compared the global P value (log-rank), Akaike information criterion (AIC), and Concordance Index (C-Index) of the two cases to determine which data format was more suitable for the age variable.

## Sample preparation for proteomics

FFPE slides were prepared by pressure cycling technology (PCT)[43,44]. Briefly, the slides were dewaxed, rehydrated, and de-crosslinked using heptane, three different concentrations of ethanol (100%, 90%, and 75%), and 100 mM tris-base solution (pH = 10), respectively. Next, the samples were lysed using PCT with a buffer containing 6 M urea, 2 M thiourea, 10 mM tris (2-carboxyethyl) phosphine, and 40 mM iodoacetamide. Then, the samples were digested using trypsin and lysC. Finally, the digested peptides were desalted by C18 (SOLAμ columns, Thermo Fisher Scientific, USA). The chemicals were bought from Sigma-Aldrich (USA), and the enzymes were obtained from Hualishi Scientific (Beijing, China).

Cleaned peptides were labeled using TMTpro 16-plex reagents (Thermo Fisher Scientific, USA). Each batch comprised 15 samples and one pooled sample, which were separated into 30 fractions within a 60 min gradient on Ultimate Dinex 3000 (Thermo Fisher Scientific, USA) equipped with a C18 column (300 Å, 5 μm × 4.6 mm × 250 mm, XBridge Peptide BEH, Waters, USA).

## Proteomics data acquisition

Each fraction was analyzed using liquid chromatography-mass spectrometry (nanoflow DIONEX UltiMate 3000 RSLCnano System and Orbitrap Exploris 480 with FAIMS Pro™, Thermo Fisher Scientific, USA). In each acquisition, peptides were separated using a 60 min gradient (from 3% to 28% buffer B (98% acetonitrile (ACN) and 0.1% formic acid)) at a 300 nL/min flowrate on an analytical column (1.9 μm 100 Å C18-Aqua, 150 mm × 75 μm). Buffer A was composed of 2% ACN, 98% $H_2O$, and 0.1% formic acid. All reagents were mass spectrometry-grade. The mass-to-charge (m/z) range of the MS1 was 375–1800 Th with a resolution of 60,000 full widths at half maximum (FWHM); the MS2 resolution was 30,000 FWHM. The turbo-TMT and advanced peak determination were enabled.

## Proteomics data analysis

Proteomic raw files were searched using Proteome Discoverer (v2.4.1.15) against a FASTA file containing 20,368 entries (human Swiss-Prot database). Channel TMT-126 was set as the reference for each batch. Correction factors (Lot# VG306794) were used when we did the database searching. The search parameters were set as follows: two trypsin missed cleavages allowed; minimal peptide length of six amino acid residues; precursor ion mass tolerance of 10 ppm; fragment mass tolerance of 0.02 Da. Normalization was performed against the total peptide amount. The false discovery rate thresholds were set to strict 1% for peptide and protein identification and quantification. Other settings were left to their default values.

## Proteomics data quality control and preprocessing

The data quality was assessed by evaluating the coefficient of variation (CV) across the pooled samples and the technical replicates. When calculating CVs, the missing values were omitted, and log2-transformed protein abundance was used.

The R package *NAguideR* was used for missing value imputation, and the impseqrob method (for robust sequential imputation) was used. Next, the batch effects correction of the resulting protein matrix was performed using Combat, an empirical Bayes framework from the R package *sva*[45]. For the matrices after imputation and correction, the non-positive values in the matrix were replaced by half the minimum value of the positive abundances of the corresponding protein. Each pair of technical replicates were then combined to one sample by calculating the mean protein abundance.

The differentially expressed proteins (DEPs) were identified with fold change (FC) values to be greater than 1.2 or 1.5 (1.2 for modeling and 1.5 for enrichment analysis), with an adjusted Welch's $t$ test $P < 0.05$. To avoid losing too many proteins by simple filters like FC, we adopted protein list from FC threshold 1.2 to give our model more freedom to decide by itself (though more time-consuming) which protein feature to use. In the enrichment analysis, to avoid over-complicated results without specificity caused by too many protein inputs, we used a strict threshold of FC 1.5.

## Tumor immune microenvironment analysis

CIBERSORTx[12] (https://cibersortx.stanford.edu/) was utilized to profile the proportions of seven immune cell types in our proteomic data. The software required a feature matrix which contained the gene expression profiles of each cell type of interest. We used a custom signature matrix generated from published thyroid cancer single-cell RNA data[46]. The asterisks marked significant difference between PB and PM by two-sided Welch's $t$ test.

## The analysis of immune checkpoints

Thirty-one immune checkpoint proteins were quantified in our data. We conducted two-sided Welch's $t$ test to compare the abundance of these immune checkpoint proteins between PB and PM using R (v4.1.1). Samples with extreme values defined by Tukey's fences were removed before plotting the boxplots.

## Multiplex immunohistochemistry staining and image analysis

Multiplex immunohistochemistry (mIHC) staining was performed using methods and reagents following the TSA Opal mIHC protocols (Akoya Biosciences/PerkinElmer). Briefly, 5 μm thickness FFPE tumor sections were stained with DAPI and antibodies against the following markers: CD3 (cat# ab135372, dilution 1:500, Abcam), CD4 (cat# ab288724, dilution 1:1000, Abcam), and CD8 (cat# ab17147, dilution 1:500, Abcam). All markers were sequentially applied and stained using their respective fluorophores in the Opal 7 kit (cat# NEL797001KT; Akoya Biosciences/PerkinElmer). Stained slides were scanned using the multispectral microscope, Vectra v3.0.3 imaging system (Akoya Biosciences/PerkinElmer), under fluorescence and low magnification at $10 \times 40$. Following scanning, around four regions of interest (each region of interest (ROI) 0.6522 mm$^2$) were selected per sample using the phenochart viewer v1.0.9 (Akoya Biosciences/PerkinElmer). ROIs were analyzed by the image analysis software, InForm v2.8.2 (Akoya Biosciences/PerkinElmer).

## Predicting PPTC recurrence risk using clinical or/and protein features

To build models for predicting PPTC recurrence risk, the PM samples were randomly divided into a training set ($n = 50$, ~60%) and an independent test set ($n = 35$, ~40%). The training set was used for building prognostic prediction models, including hyperparameter tuning, feature selection, and model training. The independent test set was used to evaluate our models' generalization capability.

We built five models using the R package *mlr3*. Specifically, we generated a Cox proportional hazard model based on clinical features (CliCox), a random survival forest based on clinical features (CliRsf), a Cox proportional hazard model based on protein features (ProtCox), a random survival forest based on protein features (ProtRsf), and a random survival forest based on clinical and protein features (CliProtRsf). For each model, we tuned the hyperparameters using grid search strategy and threefold cross-validation, selected the features, and trained the model using the training set. Lastly, we compared the C-Indexes of the five models in training, cross-validation, and test sets. The models based on the eleven clinical features did not conduct the feature selection step due to the small number of clinical features, and for the ProtCox model, we used the Least Absolute Shrinkage and Selection Operator (LASSO) for selecting the protein features. As for ProtRsf and CliProtRsf, we made the feature selection as described next. Hyperparameters were first optimized, and then 1548 DEPs (PB vs. PM; FC > 1.2, adjusted $P < 0.05$) were used for feature selection. Clinical features were also used besides DEPs in the case of CliProtRsf. The models were trained for 100 times with different initial states. In each training, we ranked the features according to permutation importance and selected the 50 most important features. Finally, we recorded the selected numbers of each protein and chose the features selected no less than 50 times as the final feature set.

## Prognostic stratification of the PM patients

Using our previously developed ProtRsf model, we predicted the prognostic survival curve of each PM patient. Then, according to the prognostic curves, the expectations corresponding to these curves were calculated and used to compute the recurrence risk score, noted as Continuous risk ranking (Crank), which is proportional to the recurrence risk. Next, using the training set, we chose the stratification threshold using the Crank scores from the recurrence and the non-recurrence groups. Specifically, the threshold was calculated by averaging the mean Cranks of two groups. We then classified the PM samples as high or low-risk using this threshold. Finally, we validated our stratification threshold using the independent test cohort, which allowed us to evaluate the generalization ability of the final model.

## Bioinformatics and statistical analyses

Statistical analysis was conducted using R (v4.1.1) and SPSS (v 23.0). The Uniform Manifold Approximation and Projection (UMAP) visualization was performed using the R package *UMAP*. The heatmap was generated using the R package *pheatmap*, with protein-level normalization and hierarchical clustering (Euclidean distance and complete option were used). The data for normality was determined by Shapiro–Wilk's test. The $P$ values of the DEPs in the volcano plots were derived from a two-sided unpaired Welch's $t$ test and adjusted using the Benjamini–Hochberg method. Pathways and networks were analyzed using the Ingenuity Pathway Analysis (IPA) and visualized with Cytoscape (v3.8.2). Gene ontology enrichment analysis was conducted by enrichGO function in R Package *clusterProfiler* (v4.0.5) using database org.Hs.eg.db (v3.13.0, stored in R package *org.Hs.eg.db*). Log-rank test was used for comparing Kaplan–Meier curves in two sample groups. For the tables of clinical characteristics, continuous variables were reported as mean ± standard deviation (SD), and categorical variables as frequency and proportion. Two-sided Wilcoxon rank-sum test (for continuous variables) and chi-squared test (for categorical variables) were used for comparison. The Pearson correlations between the fractions of immune cells and the abundance of the selected 19 proteomic features were calculated using the R package *Hmisc*.

**Reporting summary**

Further information on research design is available in the Nature Portfolio Reporting Summary linked to this article.

## Data availability

The mass spectrometry proteomic raw data generated in this study have been deposited to the ProteomeXchange Consortium via the iProX partner repository under accession identifier IPX0006407000 (subproject ID: IPX0006407001) or https://proteomecentral. proteomexchange.org/cgi/GetDataset?ID = PXD050347]. Raw data and processed data essential to this work are provided in the Supplementary Information and Source Data file. Source data are provided with this paper.

## Code availability

Code for statistical analysis, modeling and visualization presented in this manuscript and generating corresponding figure panels and tables is publicly available on Zenodo at https://zenodo.org/records/10730561.

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

## Acknowledgements

We thank Shanjun Chen from Westlake Omics Inc. for his help with improving the text. We also thank for the assistance in data storage, computation, and peptide fractionation by the Westlake University Supercomputer Center and the Mass Spectrometry & Metabolomics Core Facility at the Center for Biomedical Research Core Facilities of Westlake University. Figure 1a was created through Biorender.com. This work was supported by the National Key R&D Program of China (No. 2021YFA1301602, 2021YFA1301601, 2020YFE0202200 to Tiannan Guo), China Postdoctoral Science Foundation (2022M722841 to Yaoting Sun), and National Key R&D Program of China (No. 2022YFF0608403 to Yi Zhu). This work was further supported by the Science and Technology Project of Shenyang City (21-173-9-31 to Hao Zhang) and Applied Basic Research Program of Liaoning Province (2022020225-JH2/1013 to Hao Zhang).

## Author contributions

T.G., H.Z., Z.W. and Y.S. designed the study. C.W. prepared samples for proteomics data. T.H. and L.T. conducted database searching. H. Wang conducted data analysis. Y.Zhou conducted tumor immune microenvironment and checkpoints analysis. H.Wu processed the multiplexed immunofluorescent staining and data interpretation. Y. Zhu, M.L. and L.L. helped improve the manuscripts.

## Competing interests
