## [Peer Review File · Nature Communications]

An individualized protein-based prognostic model to stratify pediatric patients with papillary thyroid carcinomaReviewers' Comments:

Reviewer #1:

Remarks to the Author:

Wang et al. have done a good job characterizing a large cohort of patients with papillary thyroid carcinoma. A major finding is development of a prognostic prediction model for pediatric papillary thyroid carcinoma. The results are important and could be publishable, but the reviewer has some concerns that need to be addressed first.

Line 29: 'Pediatric papillary thyroid carcinomas (PPTCs) are with high inter-tumor heterogeneity...' The wording here is a bit awkward. Consider rephrasing to something like "Pediatric papillary thyroid carcinomas (PPTCs) exhibit high inter-tumor heterogeneity..."

Line 158: 1.2 fold change is a very, very low threshold. Authors need to justify its use here. Although these are statistically significant values based on adjusted p-value, how biologically relevant are they? Do the pathway enrichment (or modeling results) change significantly if a 1.5 FC threshold is used instead? IPA is mentioned elsewhere, but it is generally unclear what pathway enrichment software/database is being used to generate the results.

Line 163: 'According to the enrichment analysis of annotated keywords performed using STRING database, the most upregulated proteins in PM, compared to the other two groups, were involved in MHC-II and immunity.' Please provide the enrichment analysis results in the manuscript (or in supplemental). Is it surprising that all of the top pathway results are immune related? Were there other pathway results with better p-values that were not shown? Are these pathway hits p-values or adjusted p-values? Are these still significant when adjusted p-values are considered?

Line 168: What proteins are used for the enrichment? The 1.5 fold change or the 1.2 fold change? Please state in the text. Pathway enrichment results should be provided in supplemental materials.

Line 179: Authors should justify use of a gene deconvolution tool CIBERSORTx on proteomics data and provide evidence that this is working as expected. Immune cell deconvolution based on gene signatures requires dozens if not hundreds of genes. How good is the coverage of the genes in gene signatures given this is proteomics data? Are these Welch's t-test p-values? Is the condition of normality satisfied for using a t-test here? Do the results/significant differences change if a non-parametric test like Wilcoxon rank-sum test is used?

Line 222: Given the prominence of immune-related findings earlier, is it surprising that there were not more immune related hits here? Were any of these proteins also differentially expressed? Do these proteins belong to related pathways or families?

Line 237: 'ITGA4 and GAL3ST4 were found positively correlated to CD8+ T cells in both groups.' Whenever a quantitative statement is made, the value and corresponding p-value should be provided. Please fix here and throughout.

Line 231: Wilcoxon did not seem to be mentioned in the methods. Are these Wilcoxon rank-sum tests, specifically? Please be more specific.

Line 242: Are there any independent cohorts with protein expression (or even gene expression) for validating the signature? Translational impact of the manuscript could be much higher if the signature can be validated with independent samples (or experiments). How well does the signature work in adults? Could TCGA gene expression data from (adult?) thyroid cancers be used as some form of validation?

Line 419: More PD parameters are required to enable replication of results. Did authors use TMT

channel bleed-over correction setting in PD?

Line 430: Did authors follow internal reference scaling (IRS) to remove plex-to-plex batch effects? If not, then how were the pools used? Anecdotally, IRS seems to outperform ComBat for removing plex-to-plex batch effects. Authors should consider trying this method to see if it improves differential expression results/model prediction.

Line 434: Welch's t-test is used line but then Student's t-test is mentioned line 445-446. Why not use Welch's t-test throughout? Why assume equal variances here for these checkpoint proteins?

Line 431: Although the ComBat approach does result in negative abundances, the reviewer thought this was okay because the transformed values are changed into some other arbitrary units. If these are being treated as relative abundances, then the sign of the value might not matter as much as the differences in abundances across groups. If having negative values is okay, then arbitrarily replacing negative values can alter the distribution of the transformed data and adversely impact the downstream analyses. Please provide additional justification for handling the negative values or please change how the data is processed.

Lines 433-434: The differentially expressed proteins (DEPs) were identified with fold change (FC) values to be greater than 1.2 or 1.5 (for different purposes)' What different purposes? Please explicitly state here in methods so readers do not need to search through all of the results.

Lines 441-442: 'We used a custom signature matrix generated from published thyroid cancer single-cell RNA data'. Given the disconnect between gene expression and protein expression (and even the disconnect between single cell gene expression and bulk gene expression), please justify the use of this custom signature matrix. Has this been previously validated in other proteomic datasets?

Line 481: Information about feature selection for the modeling should be briefly included in results too.

Line 517: Identifier IPX0006407000 did not return any results when searched on iProX. The reviewer assumed <https://www.iprox.cn/> is the correct website, but more information and possibly a doi/url should be provided. Although it is deposited in iProX, will it be indexed by the ProteomeXchange Consortium? Please make sure data is available.

Authors should provide all protein abundance matrices, pathway enrichment results, etc. as part of supplemental materials.

Authors filled out a form with a link to a github repository that was empty. Please make sure code is available that can be used to reproduce the modeling results. A link to the working github repository should be included in the manuscript.

Line 722: IHC images are very small. Please make them bigger. Does the 50um scale bar apply to all images? The scale bar should be included in all images or this should be directly stated in the text.

Reviewer #2:

Remarks to the Author:

Wang et al's manuscript is an interesting and novel approach to create a clinical path to predicting the risk of recurrence in pediatric patients diagnosed with papillary thyroid cancer. As the authors accurately state, a 'one-size-fits-all' treatment strategy is associated with the potential for over-treatment of patients with low-invasive disease and, potentially, under-treatment of patients with invasive disease.

The current commercially available, molecular diagnostic panels used to augment cytological data are based on detection of somatic oncogenic driver alterations. Unfortunately, within each oncogenic driver, there is significant variability in the extent of invasive disease and response to therapy where additional, multi-omic data, including proteomic data, may provide further insight in predicting clinical behavior of the tumor.

Wang et al's proposed manuscript provides a first look at how proteomics may be used to predict the risk of recurrent disease in pediatric patients with PTC. The established expertise of the research group, depth of proteomic analysis, the analysis of five different models for data analysis, and assessment of the 19-protein model panel to analyze recurrent risk for high- versus low-risk patients. The stated limitation of the study is the lack of investigation of fine needle aspiration samples with prospective clinical follow-up to validate the reliability and accuracy of the proposed 19-protein panel model.

Questions

1) The authors report that patients were followed with cervical neck ultrasound and thyroid labs every 3 to 6 months after initial surgery, with increasing time between surveillance if patients had no evidence of persistent disease (Methods, line 355). The authors define remission as two consecutive, negative WBS and ultrasound with Tg and TgAb in the 'ideal range' (line 357-359). The authors define recurrence as structural or biochemical, with the latter defined by unstimulated Tg > 1 ng/mL, stimulated Tg > 10 ng/mL or increasing TgAb levels (line 359-364). These definitions are critical to analyzing the data as they form the basis of building and interpreting the proteomic data and model.

Of the 85 pediatric patients with PTC, the authors report that 47 (55.29%) underwent lobectomy and 38 (44.72%) underwent total thyroidectomy (Results, line 95). All patients underwent prophylactic central neck dissection (Results, line 96) and patients that underwent total thyroidectomy received RAI therapy (Methods line 353).

- \ The text currently states the definition of biochemical remission of Tg and TgAb as levels in the "ideal range". Is the "ideal range" for Tg and TgAb used to define remission the same as the for recurrence?
- \ Can the authors explain how remission and recurrence was defined for patients that underwent lobectomy? The Tg and TgAb definitions provided in lines 359-364 would be consistent for patients that underwent total thyroidectomy + RAI but cannot apply to patients that underwent lobectomy.
- \ For the patients that had recurrent disease (n = 10, Results, line 245), how many had undergone lobectomy versus total thyroidectomy + RAI?
- \ How did the authors incorporate the differences in reliability and accuracy of remission for lobectomy vs total thyroidectomy + RAI into the model predicting recurrent disease?
- \ Is there any differences, improvement or decrement, in the predictive accuracy of the model if the authors only include patients that received total thyroidectomy and RAI?

2) In previous studies, both the PTC variant/subtype as well as the somatic oncogenic alteration have been shown to correlate with invasive behavior, including the risk for regional as well as distant metastasis. Patients that present with unifocal, encapsulated thyroid cancer, associated with low-invasive oncogenes (PTEN, DICER1, RAS) have a low risk for metastasis and a high likelihood of stable remission. In contrast, patients presenting with tumors having invasive/lobulated margins and high-risk oncogenes (i.e., BRAF V600E and fusions; PMID 35015563) have an increased rate for both regional and distant metastasis with the most common, associated PTC variants being classic PTC, diffuse sclerosing variant PTC, and widely invasive follicular variant PTC.

The authors report exclusion of PTC 'subtypes with highly invasive disease, i.e., tall-cell variant, columnar and PDTC' (Methods, line 344) and do not provide any somatic oncogene data. However, the cohort has a nice variance in regard to the breadth of regional metastasis, with 82% having LN

metastasis, 31% (n = 26) with N1a and 51% (n = 43) with N1b disease. Only 1 patient with M1 disease is included (Supplement table 1).

\ What PTC subtypes are included in the study? How many with cPTC? Diffuse sclerosing PTC, and widely-invasive follicular variant PTC?

\ What is the somatic genotype for the PTC tumors included in the analysis and did the oncogenic driver correlate with the model predicting persistent/recurrent disease? i.e. is there a clinical advantage to using the proposed 19-gene model over currently available oncogenic driver panels?

\ If proteomics is brought into clinical practice, what is the estimate cost difference between a comprehensive, somatic oncogene panel vs. the 19-protein model panel from this study?

3)Minor

\ Line 64 -> reference #5 is not applicable as this reference examines radiation induced PTC (excluded from the study cohort)

\ Line 70 -> references 8-10 are very outdated and BRAF is reported in about 40% of pediatric PTC. Consider replacing with reference #6 as well as PMID 35015563).

4)As the authors suggest, a multi-center, prospective study in pediatric patients using FNA is needed to validate the 18-protein panel. It would also be interesting to test the panel's ability to predict patients that present with M1 disease.

5)There is increasing data on how to stratify surgery and surveillance based on the sonographic features of thyroid tumors and cervical neck lymph nodes with somatic oncogene driver and gene sequencing. The burden on the authors is to show that this 18-protein panel provides a more reliable and accurate tool in identifying patients at risk for recurrent disease above the current standard of care. If the panel is validated for FNA, do the authors envision this as an additional, complimentary tool to somatic oncogene analysis?

Reviewer #3:

Remarks to the Author:

In this work, Wang et al. performed detailed individualized protein-based prognostic model to study papillary thyroid in pediatric benign, pediatric malignant and adult malignant individual. Wang et al. study used machine learning-based algorithm to predict classifier contribution to tumour recurrence and simultaneously stratified PPTC patients into high and low recurrence risk group. The experimental design is good and the statistical control in all phases is appropriate. However, I consider that some major & minor points along the manuscript should be addressed:

Comments

The manuscript sections should be reorganized to facilitate the understanding and reading

Study population and samples collection

a. The clinical samples were collected between November 2007 and April 2021 for around 15.5 years. The IEC / IBSC Study number with First Hospital of China Medical University is missing.

b. Line No. 356 After the surgery, the patients were followed up every 3-6 months with cervical ultrasounds and thyroid functional examinations. In an average duration of 15.5 years how many times were these patients followed with a magnitude of (follow-up of 3-6 months).

c. Line no. 357- Re-examination was prolonged for patients with negative ultrasounds or CT, low serum thyroglobulin level, or no persistent disease (How many patients).

d. Line360- 3 Two types of recurrence, Structural recurrence based on imaging techniques and Biochemical recurrence based on clinical parameters, were studied for Cox proportional hazard (CoxPH) model with combined prognosis information: recurrence events, the time interval between surgery and recurrence, or between surgery and the last follow-up.

Each Clinical features identified the factors whose P values were less than 0.05, were taken forward. In a holistic view how many features (10 clinical features) were significantly, uniformly had a

probability range of 0 to 1 in a combined analysis.

e. There should be a paragraph titled 'Experimental Design and Statistical Rationale' that describes the number of biological and technical replicates analyzed and justification for how this provides statistical significance for the results reported. The information is scattered and need to be put together

Proteomics data acquisition and data analysis

Around 240 thyroid nodules were collected from 234 patients, the FFPE samples were processed by the protocol optimized by Zhu Y et al and Nie X et. al, cleaned peptides were labelled with TMTpro 16-plex reagents and analysed in DIONEX UltiMate 3000 RSLCnano System and Orbitrap Exploris 480 with FAIMS Pro™. The data were analysed using Proteome Discoverer (v2.4.1.15).

Tumour microenvironment was then studied with CIBERSORTx, which led to the identification of 7 immune cell types which is further analysed for immune check points. mIHC Staining of FFPE tumour sections were also performed for mainly CD3, CD4 and CD8.

a. However during the Proteome Discoverer analysis, the ability to account for reporter ion isotopic impurities for TMT10plex reagents (correction factors while designing the study); implementation of TMT quantification based on S/N values was not mention

b. Have the authors also accounted for the TMT labelling efficiency

c. Would like to know whether its FC or Log2FC, FC difference of 1.2 is very less.

d. How much peptide was injected in column

Predicting PPTC recurrence risk, stratification

PPTC was predicted with training data set of 60% and independent test data set of 40% using Cox proportional hazard model, random survival forest on both clinical and protein features. The training data set was utilized in hyperparameter tuning, feature selection, and model training and independent test set was used to evaluate our models. 3-fold cross-validation, selected the features, and trained the model using the training set. In the ProtCox model LASSO was used for selecting the protein features. 1,548 DEPs (PB versus PM; FC > 1.2, adjusted P < 0.05) were accompanied by clinical features in the case of CliProtRsf.

ProtRsf model was used to build prognostic survival curve of each patients resulting in Continuous risk ranking (Crank) and considered as recurrence risk. Crank scores were further used to stratify recurrence and the non-recurrence groups using threshold by averaging the mean Cranks of two groups. These threshold was further using the independent test cohort.

Results

Line number 81- 109 (The overall study design.....during the postoperative follow-up) can trimmed down to 1 paragraph and most of the data must be included in Study population and samples collection section

Three clinical features are the risk factors of PPTC recurrence

Univariate Cox proportional hazard (CoxPH) model showed that age, TLNN, and LLNN may be risk factors for recurrence in pediatric patients. To determine the form of the age variable, the ten clinical features were next used as the inputs of multivariate CoxPH models.

Protein qualification

Around 1272 proteins was removed with a missing value (NA) rate above 85%. CV of proteins were calculated cross the 16 pooled samples were mainly between 0.0 and 0.2, with a median of 0.0493 the CVs of the proteins across each pair of replicates were mostly less than 0.2, with medians of 0.0662, 0.0947, 0.1238, 0.0890, 0.0645 and 0.1123.

- Would recommend plotting correlation plot in which one half will be representing the intra sample correlation and other half inter sample correlation, thus one can determine the intra and inter sample correlation of individual sample.

- One should mentioned the dimensionalities of data processing in terms of Normalization, Scaling and transformation

Proteomic differences among pediatric malignant, pediatric benign and adult malignant thyroid nodules

Around 243 was significant in (PM vs. PB) and 121 proteins were significant in (PM vs. AM) differentially expressed proteins (DEPs) with fold change (FC) > 1.5 and adjusted P < 0.05.

- Again the axis of Figure 3A, 3B and 3C are showing $-\log_{10}$ (Abundances of P value) vs $\log_2(\text{FC})$. Thus during the volcano plot (FC) or $\pm \log_2(\text{FC})$ was considered was not exclusive

- One should mention this DEPs are both significant and differentially expressed as its fold change (FC) $\geq \pm 1.5$ and adjusted $P < 0.05$.

By lowering our FC threshold to 1.2, the number of DEPs increased to 1548 (PM vs. PB) and 1629 (PM vs. AM)

- What was the reason of lowering the FC, these are from FFPE tissue samples, and proteins must be significantly dysregulated as exceptionally observed for autoantibodies in serum/plasma samples.

- Authors have performed a differential proteome analysis between (PM vs. AM) and (PM vs. PB) . To clarify the message for the scientific community, a functional clustering of differential expressed proteins between (PM vs. AM) and (PM vs. PB) is needed to be elaborated in main figure. This figure should complement the actual figure 3. Currently, this information is present in fragmented, volcano plots can be moved to figure S2A. The molecular functions their ORA or GSEA and their enrichment scores can be brought in Figure 3.

Immune infiltration and expression level of immune checkpoints in pediatric thyroid nodules

In-Silico tumour microenviron were studied using 'in-silico flow cytometry' CIBERSORTx. Seven types of immune cells were imputed namely CD8+ T cells ($P = 3.7 \times 10^{-12}$), macrophages ($P = 0.031$), dendritic cells ($P = 1.4 \times 10^{-5}$) and Treg cells ($P = 0.007$). CD8+ T cells and macrophages are increased in PM samples, while dendritic cells and Treg cells are reduced in PM samples. CD8+ T cells and macrophages are increased in PM samples, while dendritic cells and Treg cells are reduced in PM samples which were validated using immunofluorescent staining.

Analysis of 19 feature proteins

Random survival forest algorithm selected 19 proteins as features for the ProtRsf model, network analysis showed that 13 of the 19 protein features were directly or indirectly connected to disease.

- What about protein interactomes specifically modulated in each cohort comparison? Authors should complement their analysis including protein interactome networks, looking for specific hubs in each cohort comparison. To enrich the discussion section, are there specific transcription factors or upstream regulators that might be involved in the downstream modulation observed at proteome level in each cohort of PPTC comparison?

- Authors can also perform targeted proteomic validation on selected protein features using SRM/PRM to see the trend of this proteins across each cohort of PPTC comparison.

Point-by-point response to the reviewers' comments

We thank all reviewers for their thorough, constructive and positive comments on our manuscript (NCOMMS-23-24388) entitled "*An individualized protein-based prognostic model to stratify pediatric patients with papillary thyroid carcinoma*". We believe that in the revised version, all points are now addressed, further strengthening our manuscript. In a nutshell, we included additional analyses, refined figures and text passages according to the suggestions. In the pages below, each of the reviewers' comments are addressed in more detail. We provide data directly in those cases where it was not appropriate to integrate into the revised manuscript.

Reviewer #1 (comments to the author):

Wang et al. have done a good job characterizing a large cohort of patients with papillary thyroid carcinoma. A major finding is development of a prognostic prediction model for pediatric papillary thyroid carcinoma. The results are important and could be publishable, but the reviewer has some concerns that need to be addressed first.

Reply:

We appreciate the positive summary of our manuscript. In the revised version, we provided the details of our data analysis, carefully checked the data and modified our manuscript according to the suggestions.

Line 29: 'Pediatric papillary thyroid carcinomas (PPTCs) are with high inter-tumor heterogeneity...' The wording here is a bit awkward. Consider rephrasing to something like "Pediatric papillary thyroid carcinomas (PPTCs) exhibit high inter-tumor heterogeneity..."

Reply:

Thank you for the suggestion. We agree it will flow better and more clearly conveys the intended meaning with rephrasing. This sentence has been revised in the manuscript as follows: "*Pediatric papillary thyroid carcinomas (PPTCs) exhibit high inter-tumor ...*".

Line 158: 1.2 fold change is a very, very low threshold. Authors need to justify its use here. Although these are statistically significant values based on adjusted p-value, how biologically relevant are they? Do the pathway enrichment (or modeling results) change significantly if a 1.5 FC threshold is used instead? IPA is mentioned elsewhere, but it is generally unclear what pathway enrichment software/database is being used to generate the results.

Reply:

We apologize for the confusion. In this study, we used the proteins derived from fold change (FC) threshold 1.5 for all the enrichment analyses and FC 1.2 for the feature selection in the process of model construction. The following are the detailed considerations when we chose the threshold 1.2.

1. Justification for why we used fold change threshold 1.2

In this study, we used the tandem mass tag (TMT) labeling technique to quantify the proteome. The data we analyzed were adjusted by internal reference scaling firstly to mitigate the potential plex-to-plex batch effects, which means the protein abundances in our matrix were expression ratios rather than the intensities themselves. In each TMT 16-plex batch, there are 15 samples to be analyzed and one pooled sample. The relative protein abundance of 15 samples were represented by the ratios of the original intensities to the corresponding intensity of the pooled sample. The pooled samples are the same and act as a bridge between different batches. Therefore, compared to protein abundances measured using label-free quantification, these ratios have been compressed in terms of mean and scale. The fold changes derived from ratio expression are not as large as those from the original intensity, which is called Ratio Compression Effects¹. Thus, a lower threshold for fold change has been adopted in some published papers using TMT technique, such as log₂ fold change 0.25 (fold change 1.189207)² and fold change 1.2³.

2. Fold change choice in modeling

In the meantime, to avoid losing too many proteins by simple filters like FC, we adopted protein list from FC threshold 1.2 to give our model more freedom to decide by itself (though more time-consuming) which protein feature to use, which is a more precise way to build a more powerful model. In this way, the model would only select the most predictive protein combination. And the combinations derived from FC 1.2 threshold protein list would be much more and fully include those from FC 1.5 threshold, which leads to more (at least equally) powerful prediction model compared to that from FC 1.5 threshold.

3. Fold change choice in pathway enrichment analysis

In our manuscript, the IPA pathway and GO enrichment analyses were based on the protein list derived from fold change threshold 1.5 (including 243 proteins in PM vs. PB and 121 proteins in PM vs. AM). We used FC 1.5 as our threshold rather than 1.2 because there are too many proteins (more than 1000) in the protein list derived from FC threshold 1.2 and thus too many pathways (or biological processes, molecular functions, cellular components) would be enriched with high significance, leading to over-complicated results without specificity. And the enrichment results from 1.2 and 1.5 FC proteins are the same for the most significant changes, which remain to be related to immune system dysfunction.

Finally, we supplemented the enrichment methods we used for each panel in both figure legends and methods sections.

"Pathways and networks were analyzed using the Ingenuity Pathway Analysis (IPA) and visualized with Cytoscape (v3.8.2). GO enrichment analysis was conducted by enrichGO function in R Package clusterProfiler (v4.0.5) using database org.Hs.eg.db (v3.13.0, stored in R package org.Hs.eg.db)."

Line 163: 'According to the enrichment analysis of annotated keywords performed using STRING database, the most upregulated proteins in PM, compared to the other two groups, were involved in MHC-II and immunity.' Please provide the enrichment analysis results in the manuscript (or in supplemental). Is it surprising that all of the top pathway results are immune related? Were there other pathway results with better p-values that were not shown? Are these pathway hits p-values or adjusted p-values? Are these still significant when adjusted p-values are considered?

Reply:

Thank you for the suggestions. We have provided the details in the supplemental files. Results from the STRING database are in the **Supplementary Table 4**. These immune-related results shown in our manuscript are all based on significantly dysregulated pathways or function enrichments. In order to confirm the reliability of these results, we used a variety of enrichment software or methods, *i.e.*, IPA, clusterProfiler, STRING, METASCAPE, and all of which pointed to immune system-related, a result that was indeed very surprising to us as well. This is why we chose to further increase the analysis of immune cell infiltration in **Figure 4**.

All pathways we present were not artificially selected or filtered. We used *P* value rather than adjusted *P* values in our manuscript. According to your suggestion, we also tried BH-adjusted *P* values as the cutoff; the pathways we presented still remained.

Line 168: What proteins are used for the enrichment? The 1.5 fold change or the 1.2 fold change? Please state in the text. Pathway enrichment results should be provided in supplemental materials.

Reply:

We used a 1.5-fold change as the cutoff. The two DEPs lists are added in **Supplementary Table 3** and detailed enriched pathways and functions are listed in **Supplementary Table 5**.

Line 179: Authors should justify use of a gene deconvolution tool CIBERSORTx on proteomics data and provide evidence that this is working as expected. Immune cell deconvolution based on

gene signatures requires dozens if not hundreds of genes. How good is the coverage of the genes in gene signatures given this is proteomics data? Are these Welch's t-test p-values? Is the condition of normality satisfied for using a t-test here? Do the results/significant differences change if a non-parametric test like Wilcoxon rank-sum test is used?

Reply:

Since multiple dysregulated immune-related pathways and biological processes were enriched, we would like to further explore the difference of immune microenvironment in PB and PM samples. Thus, we conducted the immune infiltration analysis using CIBERSORTx. Our customized gene signature matrix contains 2223 genes, which is derived from the published RNAseq data of thyroid papillary carcinoma. Among the 2224 gene features, 1451 (65.3%) genes can be matched to the protein abundances, which is higher than the 50% threshold suggested by the software. The authors of CIBERSORTx have proposed possible estimation biases introduced by inter-platform variations. They have provided a batch correction method as an optional parameter in CIBERSORTx to deal with this problem, and the deconvolution results substantially improved compared to the ground truth⁴. We applied the batch correction provided in CIBERSORTx to our analysis to reduce the influence of inter-platform bias. Also, CIBERSORTx has been used in many proteomic studies⁵⁻⁷.

In our submitted manuscript, we used Welch's *t*-test, which was done by the `stat_compare_means` function in R package *ggpubr*. We tested the data for normality using Shapiro-Wilk's test and found that out of the 14 sets of data (7 cell types * 2 datasets), 6 sets of data did not fit the normal distribution and 8 sets did. If the Wilcoxon Rank-Sum Test is used instead, the results are shown below. $P < 0.05$ for all cell types except CD4+ T cells, but only CD8+ T cells and dendritic cells meet $P < 0.01$.

The statistical analysis method has been added in the revised legend of **Figure 4**.

Legend: Relative proportions of seven types of immune cells in PB and PM samples imputed by CIBERSORTx. The significance was determined by Wilcoxon Rank-Sum Test. (Not used in our manuscript)

Line 222: Given the prominence of immune-related findings earlier, is it surprising that there were not more immune related hits here? Were any of these proteins also differentially expressed? Do these proteins belong to related pathways or families?

Reply:

We think that is not surprising. These two lists of proteins represent two aspects. The dysregulated proteins related to immune systems are from the differences between benign and malignant pediatric thyroid nodules. However, proteins mentioned in Line 222 evolved from the differences between pediatric papillary thyroid cancers with recurrence and un-recurrence. To avoid data leakage, we didn't use the differentially expressed proteins (DEPs) between recurrence and un-recurrence groups to build the predicting model. Instead, we applied the DEPs from malignant vs. benign groups with 1.2 FC as the protein candidate.

We did the network analysis for the selected protein in Line 222. Our network analysis showed that 13 of the 19 protein features were directly or indirectly connected. In particular, LGALS3, the hub protein, may perform a significant role in pediatric thyroid carcinoma (**Figure 5E**).

Line 237: 'ITGA4 and GAL3ST4 were found positively correlated to CD8+ T cells in both groups.' Whenever a quantitative statement is made, the value and corresponding p-value should be provided. Please fix here and throughout.

Reply:

Thank you for your suggestion. The modified text reads as follows.

*"ITGA4 ($P = 7.28 \times 10^{-4}$ and $P = 2.05 \times 10^{-8}$ in high and low risk groups, respectively) and GAL3ST4 ($P = 1.56 \times 10^{-4}$ and $P = 1.24 \times 10^{-4}$ in high and low risk groups, respectively) were found positively correlated to CD8+ T cells in both groups. For the 31 immune checkpoint proteins quantified, only the abundance of IL10RB was found to decrease with the predicted recurrence risk and highest in PB samples ($P = 0.0012$ and 0.038 , respectively; **Supplementary Figure 3**)."*

Line 231: Wilcoxon did not seem to be mentioned in the methods. Are these Wilcoxon rank-sum tests, specifically? Please be more specific.

Reply:

Thank you for your kind reminder. For **Figure 5F**, two-sample Wilcoxon Rank-Sum Tests were used to estimate the P values. We refined statistical tests for each figure and added the names of tests legends.

Line 242: Are there any independent cohorts with protein expression (or even gene expression) for validating the signature? Translational impact of the manuscript could be much higher if the signature can be validated with independent samples (or experiments). How well does the signature work in adults? Could TCGA gene expression data from (adult?) thyroid cancers be used as some form of validation?

Reply:

We recognize that additional center and sample validation can further confirm the reliability of the proteins selected by this method. However, this study is currently a single-center cohort study, which is a limitation of this study at this time. In the future, we hope to further confirm the robustness of our biomarkers through prospective multicenter clinical trials.

There are significant differences in both protein expression and phenotype between thyroid cancers in children and adolescents and those in adults, so it is reasonable to assume that this method is not applicable to adults. Recurrent papillary thyroid cancer in adults is another project in our team. In addition, according to previous studies, it has been demonstrated that the expression at the mRNA level correlates less than 60% with the expression at the protein level, and therefore the protein information is not directly transposable to the mRNA molecular level. Therefore, if it is necessary to analyze from the mRNA level, it is necessary to construct a separate classifier. We investigated the TCGA data, and there were only 14 cases of thyroid cancer in pediatric (all of them were older than 14 years old). More importantly, there was no recurrence information provided. Thus, TCGA data is not possible to be used for validation.

Line 419: More PD parameters are required to enable replication of results. Did authors use TMT channel bleed-over correction setting in PD?

Reply:

Thank you for your advice. We added the detailed information in the methods section, as shown below.

"Proteomic raw files were searched using Proteome Discoverer (v2.4.1.15) against a FASTA file containing 20,368 entries (human Swiss-Prot database). Channel TMT-126 was set as the reference for each batch. Correction factors (Lot# VG306794) were used when we did the data searching. The search parameters were set as follows: two trypsin missed cleavages allowed; minimal peptide length of 6 amino acid residues; precursor ion mass tolerance of 10 ppm; fragment mass tolerance of 0.02 Da. Normalization was performed against the total peptide amount. The false discovery rate thresholds were set to strict 1% for peptide and protein quantification. Other settings

were left to their default values."

Line 430: Did authors follow internal reference scaling (IRS) to remove plex-to-plex batch effects? If not, then how were the pools used? Anecdotally, IRS seems to outperform ComBat for removing plex-to-plex batch effects. Authors should consider trying this method to see if it improves differential expression results/model prediction.

Reply:

Yes, we have incorporated internal reference scaling (IRS) in our analysis. Immediately after the database search, IRS was used to correct the plex-to-plex batch effects in the protein expression matrix. Specifically, we added a pooled sample in each batch, and the relative protein abundance of the other 15 samples in the same batch were calculated by dividing the intensities of the pooled sample. The pooled samples are the same among all batches, so the pooled samples acted as internal references here.

However, after quality control analysis and missing value imputation, batch effects were still detected by UMAP as shown in the below left figure. Thus, to further correct the batch effects, we applied ComBat for removing the remaining batch effects. And after sequential batch effects correction (IRS + ComBat), the batch effects were no longer significant, as shown in the below right figure or **Supplementary Figure 1E**.

UMAP show the batches of sample distribution before (left) and after (right) batch correction using ComBat.

Line 434: Welch's t-test is used line but then Student's t-test is mentioned line 445-446. Why not use Welch's t-test throughout? Why assume equal variances here for these checkpoint proteins?

Reply:

We apologize for the mistake in line 445-446. It was actually a Welch's *t*-test (the default option for *t*-test) conducted using 'stat_compare_means' function of R package *ggpubr*. Thank you again for pointing this out.

Line 431: Although the ComBat approach does result in negative abundances, the reviewer thought this was okay because the transformed values are changed into some other arbitrary units. If these are being treated as relative abundances, then the sign of the value might not matter as much as the differences in abundances across groups. If having negative values is okay, then arbitrarily replacing negative values can alter the distribution of the transformed data and adversely impact the downstream analyses. Please provide additional justification for handling the negative values or please change how the data is processed.

Reply:

Thanks for this suggestion. After internal reference scaling, the abundances in the protein matrix become the expression ratio. And these relative abundances are relative to the pooled samples, which represents how many folds of the abundances in analyzed samples when compared with the abundances in pooled samples. In this case, negative values are meaningless because we cannot expect negative folds relative to the abundance of pooled samples. Another reason is that if we did not deal with these negative values, then for some proteins, their sums across groups (*e.g.*, PB, PM, AM) would be negative, resulting in meaningless negative fold changes when doing downstream analyses. Therefore, we replaced these negative values with half the minimum

value of the positive abundances of the corresponding protein to maximumly keep the differences in abundances across groups.

Lines 433-434: The differentially expressed proteins (DEPs) were identified with fold change (FC) values to be greater than 1.2 or 1.5 (for different purposes)' What different purposes? Please explicitly state here in methods so readers do not need to search through all of the results.

Reply:

According to your suggestion, we clarify these in the method section.

"The differentially expressed proteins (DEPs) were identified with fold change (FC) values to be greater than 1.2 or 1.5 (1.2 for modeling and 1.5 for enrichment analysis), with an BH-adjusted Welch's t-test $P < 0.05$."

Lines 441-442: 'We used a custom signature matrix generated from published thyroid cancer single-cell RNA data'. Given the disconnect between gene expression and protein expression (and even the disconnect between single cell gene expression and bulk gene expression), please justify the use of this custom signature matrix. Has this been previously validated in other proteomic datasets?

Reply:

Thanks for the comment. The question is also explained in the previous comments.

To minimize the impact of cross-platform variation on the deconvolution results, there are batch-correction options provided in CIBERSORTx. The method has been demonstrated to be useful for cross-platform data⁸, for example, deconvolute cell fractions from bulk RNA-seq data using a scRNA-seq signature matrix. For proteomic data, the microarray-derived leukocyte gene signature matrix provided by CIBERSORTx, termed LM22, has been used to infer immune cell infiltration⁶. LM22 contains 547 genes that distinguish 22 human hematopoietic cell phenotypes. The reason we do not use the built-in LM22 matrix is that it is derived from peripheral blood cells and only 175/547 were identified in our matrix. Hence, we customized the signature matrix using

published scRNAseq data of papillary thyroid tumors and performed the S-mode correction which is tailored for single cell-derived signature matrices when deconvolute our data to correct the cross-platform bias. By this way, we quantified 65.3% (1451 proteins out of 2223 genes) features in thyroid thyroid-specific matrix. As for using gene expression signature matrix to deconvolute proteomics data, similar applications can be seen in many other studies^{6,7,9,10}

Line 481: Information about feature selection for the modeling should be briefly included in results too.

Reply:

Thank you for your advice. We have supplemented the related content in the Results section.

"To predict the PTC recurrence risk of patients from the PM group, the PM samples were randomly divided into a training set (n=50, ~60%) and an independent test set (n=35, ~40%). Then, we developed five models based on two algorithms (Cox proportional hazard model and random survival forest) and two types of features (clinical features and proteins). Specifically, we developed the following models: two Cox proportional hazard models based on clinical features (CliCox) or protein features (ProtCox); three random survival forests based on clinical features (CliRsf), protein features (ProtRsf), or clinical and protein features (CliProtRsf). For each model, we tuned the hyperparameters using grid search strategy and 3-fold cross-validation, selected the features, and trained the model using the training set. The final hyperparameter settings of the five models are summarized in Supplementary Table 2."

Line 517: Identifier IPX0006407000 did not return any results when searched on iProX. The reviewer assumed <https://www.iprox.cn/> is the correct website, but more information and possibly a doi/url should be provided. Although it is deposited in iProX, will it be indexed by the ProteomeXchange Consortium? Please make sure data is available.

Reply:

Yes, iProX is an integrated proteome resources center in China and is indexed by the ProteomeXchang Consortium. Please visit our data through URL:

<https://www.iprox.cn/page/DSV021.html?url=1705059006234WK3k>; Password: yhkB. All the data will be open to the public once the paper is published.

REDACTED

Authors should provide all protein abundance matrices, pathway enrichment results, etc. as part of supplemental materials.

Reply:

We have uploaded the detailed patient information table and proteome matrix (**Supplementary Table 2**), DEPs lists (**Supplementary Table 3**), and enrichment analysis results (**Supplementary Table 4 and 5**) as supplemental files.

Authors filled out a form with a link to a github repository that was empty. Please make sure code is available that can be used to reproduce the modeling results. A link to the working github repository should be included in the manuscript.

Reply:

Thanks for pointing out this. We have uploaded our codes for analysis. The repository link is <https://github.com/wanghe98/PPTC>

Line 722: IHC images are very small. Please make them bigger. Does the 50um scale bar apply to all images? The scale bar should be included in all images or this should be directly stated in the text.

Reply:

Thank you for your suggestions. The figure has been modified, and the scale bar is stated in the figure legend.

Reviewers #2 (comments to the author):

Wang et al's manuscript is an interesting and novel approach to create a clinical path to predicting the risk of recurrence in pediatric patients diagnosed with papillary thyroid cancer. As the authors accurately state, a 'one-size-fits-all' treatment strategy is associated with the potential for over-treatment of patients with low-invasive disease and, potentially, under-treatment of patients with invasive disease.

The current commercially available, molecular diagnostic panels used to augment cytological data are based on detection of somatic oncogenic driver alterations. Unfortunately, within each oncogenic driver, there is significant variability in the extent of invasive disease and response to therapy where additional, multi-omic data, including proteomic data, may provide further insight in predicting clinical behavior of the tumor.

Wang et al's proposed manuscript provides a first look at how proteomics may be used to predict the risk of recurrent disease in pediatric patients with PTC. The established expertise of the research group, depth of proteomic analysis, the analysis of five different models for data analysis, and assessment of the 19-protein model panel to analyze recurrent risk for high- versus low-risk patients. The stated limitation of the study is the lack of investigation of fine needle aspiration samples with prospective clinical follow-up to validate the reliability and accuracy of the proposed 19-protein panel model.

Reply:

We deeply appreciate the positive comments on the novel approach of our study and the acknowledgment of strengths such as our research expertise, depth of proteomic analysis, multiple modeling approaches, and focus on pediatric papillary thyroid cancer.

In response to the critique on the lack of validation with clinical follow-up, we fully agree that prospective confirmation is needed before clinical implementation. We have modified the Discussion section to note validation in an independent dataset as an important next step for future studies.

Thank you again for the thoughtful feedback, which has helped improve our work.

Questions

1) The authors report that patients were followed with cervical neck ultrasound and thyroid labs every 3 to 6 months after initial surgery, with increasing time between surveillance if patients had no evidence of persistent disease (Methods, line 355). The authors define remission as two consecutive, negative WBS and ultrasound with Tg and TgAb in the 'ideal range' (line 357-359). The authors define recurrence as structural or biochemical, with the latter defined by unstimulated Tg > 1 ng/mL, stimulated Tg > 10 ng/mL or increasing TgAb levels (line 359-364). These definitions are critical to analyzing the data as they form the basis of building and interpreting the proteomic data and model.

Of the 85 pediatric patients with PTC, the authors report that 47 (55.29%) underwent lobectomy and 38 (44.72%) underwent total thyroidectomy (Results, line 95). All patients underwent prophylactic central neck dissection (Results, line 96) and patients that underwent total thyroidectomy received RAI therapy (Methods line 353).

☐ The text currently states the definition of biochemical remission of Tg and TgAb as levels in the "ideal range". Is the "ideal range" for Tg and TgAb used to define remission the same as the for recurrence?

Reply:

We apologize for the unclear description in our manuscript.

We defined these two states according to the criteria in 2015 ATA guidelines^{11,12} copied below. Considering Tg and TgAb is not accurate for assessing recurrence in patients who underwent lobectomy, we did not apply the criteria for biochemical recurrence in the present study since some of the patients underwent lobectomy. In other words, all recurrent patients in our study are structural recurrences.

To make it clear, we modified the Methods section (deleted the biochemical recurrence).

"Disease remission and recurrence were determined according to the American Thyroid Association management guidelines^{11,12}. Disease remission was defined as two consecutive negative whole-body scans and ultrasounds."

"Due to inaccurate evaluation based on serum Tg and TgAb for patients with lobectomy, only structural recurrence was considered in this study. Disease recurrence was determined as a new disease in the thyroid bed or lymph nodes proven by cytology or histopathology, and/or confirmed by ultrasounds or CT scans, or distant metastases detected by whole-body scan."

☒ Can the authors explain how remission and recurrence was defined for patients that underwent lobectomy? The Tg and TgAb definitions provided in lines 359-364 would be consistent for patients that underwent total thyroidectomy + RAI but cannot apply to patients that underwent lobectomy.

Reply:

Patients who underwent lobectomy were also only considered for structural recurrence. The definition of recurrence for patients with different surgical procedures has been modified in the manuscript and stated above.

☒ For the patients that had recurrent disease (n = 10, Results, line 245), how many had undergone lobectomy versus total thyroidectomy + RAI?

Reply:

In our study, there are 12 patients had recurrence. Six patients underwent lobectomy, and six patients underwent total thyroidectomy, with three of them undergoing total thyroidectomy + RAI.

The ten patients mentioned in line 245 refer to the number of patients that were wrongly identified by our model, comprising two false negatives (lobectomy n=1 versus total thyroidectomy n=1) and eight false positive cases (lobectomy n=7 versus total thyroidectomy n=1).

☐ How did the authors incorporate the differences in reliability and accuracy of remission for lobectomy vs total thyroidectomy + RAI into the model predicting recurrent disease?

Reply:

This is a good question. Currently, it is controversial regarding the effect of the surgical procedure on the recurrence of pediatric PTC¹³⁻¹⁵. In our cohort, we found no significant difference in the effect of surgical approach on recurrence.

Recurrence was observed in 12.77% (6/47) and 18.42% (7/38) of patients undergoing lobectomy and total thyroidectomy, respectively. To explore the effect of the surgical approach on recurrence, we performed chi-square test, Kaplan-Meier survival analysis and Cox regression model. The chi-square test showed no significant difference ($\chi^2=0.158$, $P=0.691$), the Kaplan-Meier survival curves suggested that the difference in recurrence-free survival (RFS) between the two surgical approaches was not statistically significant ($P=0.637$), and the univariate analysis of the Cox hazard proportional regression model showed that the surgical approach was not a risk factor for recurrence in patients ($P=0.638$, HR=0.762, 95% CI:0.245-2.365). The results of multivariate CoxPH models also indicate no substantial differences. Therefore, we did not include this factor in the model to predict recurrent disease in subsequent studies.

We added the new results and revised **Figure 2B** and **2C** as shown below.

Figure 2. Analysis of the clinical recurrence risk factors for PPTC. (B-C) Forest plots for two multivariate CoxPH models using (B) continuous non-negative integer age and (C) categorical age, respectively. P values are tested by Cox proportional hazard model.

Is there any differences, improvement or decrement, in the predictive accuracy of the model if the authors only include patients that received total thyroidectomy and RAI?

Reply:

The previous analysis indicated that the surgical approach was not a risk factor for prognosis in our cohort, and there was no difference in the proportion of the number of recurrences in each group, so total thyroidectomy patients were not studied separately. Secondly, the number of patients who underwent total thyroidectomy was too small (n=38). Studying only patients received total thyroidectomy halved the study population in the experimental group and the

number of patients with observed recurrence. The decrease in cohort size was not conducive to the modeling and subsequent validation of the results. Therefore, we chose to include all pediatric patients with thyroid cancer who had either lobectomies and total thyroidectomies.

2) In previous studies, both the PTC variant/subtype as well as the somatic oncogenic alteration have been shown to correlate with invasive behavior, including the risk for regional as well as distant metastasis. Patients that present with unifocal, encapsulated thyroid cancer, associated with low-invasive oncogenes (PTEN, DICER1, RAS) have a low risk for metastasis and a high likelihood of stable remission. In contrast, patients presenting with tumors having invasive/lobulated margins and high-risk oncogenes (i.e., BRAF V600E and fusions; PMID 35015563) have an increased rate for both regional and distant metastasis with the most common, associated PTC variants being classic PTC, diffuse sclerosing variant PTC, and widely invasive follicular variant PTC.

The authors report exclusion of PTC' subtypes with highly invasive disease, i.e., tall-cell variant, columnar and PDTC' (Methods, line 344) and do not provide any somatic oncogene data.

However, the cohort has a nice variance in regard to the breadth of regional metastasis, with 82% having LN metastasis, 31% (n = 26) with N1a and 51% (n = 43) with N1b disease. Only 1 patient with M1 disease is included (Supplement table 1).

□ What PTC subtypes are included in the study? How many with cPTC? Diffuse sclerosing PTC, and widely-invasive follicular variant PTC?

Reply:

Thank you for your question. To accurately answer this question, two additional histopathologists helped to carefully revisit HE slides for pediatric PTC. If the area of the particular pathological subtype accounted for more than 30% or more of the entire section, it was determined to be the corresponding pathologic subtype¹⁶.

From the 85 patients, there are 69 cPTC (81.2%), eight diffuse sclerosing variant PTC (9.4%), six hobnail variant PTC (7.1%), one solid variant PTC (1.1%) and one columnar cell variant PTC

(1.1%). For the twelve patients with recurrence, ten of them are cPTC, one diffuse sclerosing variant PTC and one hobnail variant PTC. With the PTC subtype information, we have modified the corresponding contents in Methods section.

☒ What is the somatic genotype for the PTC tumors included in the analysis and did the oncogenic driver correlate with the model predicting persistent/recurrent disease? i.e. is there a clinical advantage to using the proposed 19-gene model over currently available oncogenic driver panels?

Reply:

We do not have information about the genetic changes in these samples for the time being. In China, genetic testing is not a routine detection, and only a small number of patients choose to be tested for the *BRAF* V600E mutation. Among these 85 samples of pediatric PTC, nine patients were tested for *BRAF*, 5 of them had *BRAF* mutations. We believe that the integration of protein information with gene information will further improve the efficacy of the model, and this is one of our future planned work.

☒ If proteomics is brought into clinical practice, what is the estimate cost difference between a comprehensive, somatic oncogene panel vs. the 19-protein model panel from this study?

Reply:

Thank you for the comments regarding the practicality of using this assay in the real world. Our study is a proof-of-principle to show that protein-based classifiers can be used to classify the recurrence risks for pediatric PTCs. Right now, we can tell that the cost to measure 19 proteins using targeted proteomics is not higher than that for measuring 19 mRNAs. However, we respectfully maintain that this is beyond the scope of this manuscript.

3)Minor

☒ Line 64 -> reference #5 is not applicable as this reference examines radiation induced PTC (excluded from the study cohort)

☒ Line 70 -> references 8-10 are very outdated and BRAF is reported in about 40% of pediatric PTC. Consider replacing with reference #6 as well as PMID 35015563).

Reply:

Thank you for pointing out these issues and recommending proper literature. We have corrected them in the revised version.

4)As the authors suggest, a multicenter, prospective study in pediatric patients using FNA is needed to validate the 18-protein panel. It would also be interesting to test the panel's ability to predict patients that present with M1 disease.

Reply:

We appreciate the reviewer's suggestions regarding validation and testing the protein panel's ability to predict metastases. As the samples with distant metastasis (M1) were limited in our current cohort (only one case), the prediction of metastatic disease represents a separate issue that requires investigation in larger patient groups in future studies.

5)There is increasing data on how to stratify surgery and surveillance based on the sonographic features of thyroid tumors and cervical neck lymph nodes with somatic oncogene driver and gene sequencing. The burden on the authors is to show that this 18-protein panel provides a more reliable and accurate tool in identifying patients at risk for recurrent disease above the current standard of care. If the panel is validated for FNA, do the authors envision this as an additional, complimentary tool to somatic oncogene analysis?

Reply:

Thank you for your comments. We believe that protein panel can be a useful tool as an additional approach in the diagnosis and evaluation of thyroid diseases. Two keywords are worth being emphasized here, one is FNA, and the other is multimodal or multilevel information. FNA is the

gold standard for preoperative diagnosis, and if it is possible to detect both genes and proteins in one FNA-biopsy sample, it can further improve the assessment of cytopathology and, at the same time, provide more information on alterations at the molecular level for risk stratification of tumors, thus guiding a more appropriate treatment. We plan to initiate larger multicenter, prospective studies in subsequent studies and validate them by means of more accurate targeted proteomics.

The second point is that multimodal data has been a hot word in recent years. Multi-dimensional information can depict the state of the tumor more comprehensively and view the tumor from different perspectives, thus obtaining a more accurate assessment, which of course cannot be separated from the support of big data and artificial intelligence. This is also one of our future endeavors.

Reviewer #3 (Remarks to the Author): expertise in proteomics of endocrine cancer

In this work, Wang et al. performed detailed individualized protein-based prognostic model to study papillary thyroid in pediatric benign, pediatric malignant and adult malignant individual. Wang et al. study used machine learning-based based algorithm to predict classifier contribution to tumour recurrence and simultaneously stratified PPTC patients into high and low recurrence risk group.

The experimental design is good and the statistical control in all phases is appropriate. However, I consider that some major & minor points along the manuscript should be addressed:

Comments

The manuscript sections should be reorganized to facilitate the understanding and reading

Reply:

Thank you for your positive feedback on our paper. We highly appreciate your comments. This helps us improve the quality of the paper and better serve the academic community and clinical practice with the research findings. The following are our point-to-point replies.

Study population and samples collection

a. The clinical samples were collected between November 2007 and April 2021 for around 15.5 years. The IEC / IBSC Study number with First Hospital of China Medical University is missing.

Reply:

Based on your comments, we made additions here with the following revisions.

"This study was approved by the Ethics Committee of the First Hospital of China Medical University with the study number 2021-287-2."

b. Line No. 356 After the surgery, the patients were followed up every 3-6 months with cervical ultrasounds and thyroid functional examinations. In an average duration of 15.5 years how many times were these patients followed with a magnitude of (follow-up of 3-6 months).

Reply:

Thank you for the questions. In this study, all the postoperative patients visited the hospital every 3-6 months for the first year after surgery. Re-examination was prolonged for patients with negative ultrasounds or CT, low serum thyroglobulin level, or no persistent disease. Patients are asked to be followed up 6-12 months in the second to fifth year after surgery and every 1-2 years after five years of surgery.

c. Line no. 357- Re-examination was prolonged for patients with negative ultrasounds or CT, low serum thyroglobulin level, or no persistent disease (How many patients).

Reply:

According to our statistics, 50 patients were followed up for more than five years, 33 patients were followed up for at least one year, and only two patients followed up for less than one year.

Follow-up (years)	Total patients	Recurrent patients	Recurrent-free patients
≤1	2	1	1
1-5	33	8	25
>5	50	3	47

d. Line360- 3 Two types of recurrence, Structural recurrence based on imaging techniques and Biochemical recurrence based on clinical parameters, were studied for Cox proportional hazard (CoxPH) model with combined prognosis information: recurrence events, the time interval between surgery and recurrence, or between surgery and the last follow-up.

Each Clinical features identified the factors whose P values were less than 0.05, were taken forward. In a holistic view how many features (10 clinical features) were significantly, uniformly had a probability range of 0 to 1 in a combined analysis.

Reply:

Among the clinical features in univariate CoxPH model, three (Age, LNN, LLNN) were significant. But in the multivariate CoxPH model, only age (treated as a binary variable) was nearly significant.

e. There should be a paragraph titled 'Experimental Design and Statistical Rationale' that describes the number of biological and technical replicates analyzed and justification for how this provides statistical significance for the results reported. The information is scattered and need to be put together

Reply:

In response to your suggestion, we have rearranged and added the following paragraph to the methods section.

"Experimental design and statistical rationale

We collected FFPE slides for proteomics data acquisition. Each slide was stained with hematoxylin and eosin and reviewed by at least two experienced histopathologists and histopathological subtypes for PM were further evaluated according to The 15th edition of the World Health Organization Classification of Endocrine and Neuroendocrine Tumors¹⁷. Each slide was reviewed and processed to make sure the tumor ratio was approximately more than 80% before proceeding to proteomic sample preparation.

We collected 240 thyroid nodules (87 PM, 85 PB, 68 AM) FFPE slides (10 μ m thickness) from 234 patients (85 PM, 83 PB, 66 AM). Two samples from each group were randomly selected as technical replicates. To minimize the potential artificial effects during experiments, we randomly allocated the 240 tissues into 16 batches. In each batch, there were 15 tissue samples and one pooled sample was used as an internal reference scaling for the batches. The replicates and pooled samples were analyzed for data quality control."

Proteomics data acquisition and data analysis

Around 240 thyroid nodules were collected from 234 patients, the FFPE samples were processed by the protocol optimized by Zhu Y et al and Nie X et. al a, cleaned peptides were labelled with TMTpro 16-plex reagents and analysed in DIONEX UltiMate 3000 RSLCnano System and Orbitrap Exploris 480 with FAIMS Pro™. The data were analysed using Proteome Discoverer (v2.4.1.15). Tumour microenvironment was then studied with CIBERSORTx, which led to the identification of 7 immune cell types which is further analysed for immune check points. mIHC Staining of FFPE tumour sections were also performed for mainly CD3, CD4 and CD8.

a. However during the Proteome Discoverer analysis, the ability to account for reporter ion isotopic impurities for TMT10plex reagents (correction factors while designing the study); implementation of TMT quantification based on S/N values was not mention

Reply:

Thank you for the advice. Correction factors (Lot# VG306794) were used when we did the data searching. The S/N value was 1.5, which is the default setting. This information is added in the revised methods section.

b. Have the authors also accounted for the TMT labelling efficiency

Reply:

Yes, we have checked the TMT labeling efficiency and then combined the labeled peptides..

c. Would like to know whether its FC or Log2FC, FC difference of 1.2 is very less.

Reply:

We apologize for the confusion. In this study, we used the proteins derived from fold change (FC) threshold 1.5 for all the enrichment analyses and FC 1.2 for the feature selection in the process of

model construction. The following are the detailed considerations when we chose the threshold 1.2.

1. Justification for why we used fold change threshold 1.2

In this study, we used the tandem mass tag (TMT) labeling technique to quantify the proteome. The data we analyzed were adjusted by internal reference scaling firstly to mitigate the potential plex-to-plex batch effects, which means the protein abundances in our matrix were expression ratios rather than the intensities themselves. In each TMT 16-plex batch, there are 15 samples to be analyzed and one pooled sample. The relative protein abundance of 15 samples were represented by the ratios of the original intensities to the corresponding intensity of the pooled sample. The pooled samples are the same and act as a bridge between different batches. Therefore, compared to protein abundances measured using label-free quantification, these ratios have been compressed in terms of mean and scale. The fold changes derived from ratio expression are not as large as those from the original intensity, which is called Ratio Compression Effects¹. Thus, a lower threshold for fold change has been adopted in some published papers using TMT technique, such as log₂ fold change 0.25 (fold change 1.189207)² and fold change 1.2³.

2. Fold change choice in modeling

In the meantime, to avoid losing too many proteins by simple filters like FC, we adopted protein list from FC threshold 1.2 to give our model more freedom to decide by itself (though more time-consuming) which protein feature to use, which is a more precise way to build a more powerful model. In this way, the model would only select the most predictive protein combination. And the combinations derived from FC 1.2 threshold protein list would be much more and fully include those from FC 1.5 threshold, which leads to more (at least equally) powerful prediction model compared to that from FC 1.5 threshold.

d. How much peptide was injected in column

Reply:

For each batch, there were 112 ug of peptides (7 ug/channel*16 channels). The peptides were further separated into 120 fractions with 120 HPLC gradient and then combined into 30 fractions (#1,#31,#61,#91 into one, and so on). For each combined fraction, we first dried them and then resolved them using 30 uL MS bufferA, and inject 4 uL peptides into HPLC-MS/MS.

Predicting PPTC recurrence risk, stratification

PPTC was predicted with training data set of 60% and independent test data set of 40% using Cox proportional hazard model, random survival forest on both clinical and protein features. The training data set was utilized in hyperparameter tuning, feature selection, and model training and independent test set was used to evaluate our models. 3-fold cross-validation, selected the features, and trained the model using the training set. In the ProtCox model LASSO was used for selecting the protein features. 1,548 DEPs (PB versus PM; FC > 1.2, adjusted P < 0.05) were accompanied by clinical features in the case of CliProtRsf.

ProtRsf model was used to build prognostic survival curve of each patients resulting in Continuous risk ranking (Crank) and considered as recurrence risk. Crank scores were further used to stratify recurrence and the non-recurrence groups using threshold by averaging the mean Cranks of two groups. These threshold was further using the independent test cohort.

Reply:

Thank you for the summary.

Results

Line number 81- 109 (The overall study design.....during the postoperative follow-up) can be trimmed down to 1 paragraph and most of the data must be included in Study population and samples collection section

Reply:

According to the advice, we modified our text as follows.

Results section

“Clinical characteristics of our study population

The overall study design is demonstrated in Figure 1A. We enrolled 85 PPTC patients (PM) and 83 pediatric patients with benign nodules (P.B.) (Figure 1B), and their clinicopathological features were summarized in Supplementary Table 1. This cohort included 23 males and 62 females (male-to-female ratio of 1:2.7) with an average age of 15.6 ± 2.4 years and 15 males and 68 females (male-to-female ratio of 1:4.5) with an average age of 15.9 ± 1.9 years in PM and PB groups respectively. All patients were admitted to the hospital with a mass in the neck, and their average tumor size was 2.4 ± 1.3 cm in PM group, which is smaller than those in PB group 3.8 ± 1.3 cm. In PM group, the median follow-up time was 71 months (interquartile range 48-113), during which no death was reported. Lung metastasis was discovered in one patient before the operation, and no change was reported after the radioactive iodine (RAI) therapy. Postoperative structural recurrence occurred in 12 cases (average age of 14): ten ipsilateral cervical lymph node metastases and one contralateral cervical lymph node metastase. One case developed postoperative lung metastases. All the cases of lymph node metastases were reoperated. During the follow-up evaluations, we found that the lesions of the patients with lung metastases had shrunk after the RAI therapy, and no growth or mental retardation was detected in any patient. Finally, no hematological or other secondary solid primary tumors were found during the postoperative follow-up.”

Methods section

“Study population

In this retrospective study, we evaluated pediatric patients (≤ 18 years) with thyroid nodules, including 85 PM and 83 PB thyroid nodules, who underwent surgery in the First Hospital of China Medical University between November 2007 and April 2021. This study was approved by the Ethics Committee of the First Hospital of China Medical University with the study number 2021-287-2.

The exclusion criteria for PM were the following: (a) with a history of radiation exposure or

family history, (b) with poorly differentiated PTC, (c) loss of follow-up or incomplete clinical data, and (d) non-primary operation. We excluded uncertain malignant potential nodules for the PB group. We also included 66 AM patients with PTC to compare pediatric and adult thyroid cancer proteomic profiling. The detailed pediatric patient characteristics are listed in the **Supplementary Table 1**.

Preoperative pulmonary computed tomography (CT) showed that one patient had multiple metastases in the lung. All patients were surgically treated. Lobectomy and ipsilateral central lymph node dissections were performed in unilateral PTC. Total thyroidectomy was performed in patients with ETE, such as the invasion of nerves, blood vessels, or trachea. Patients with bilateral PTC underwent total thyroidectomy and bilateral central lymph node dissections. For PM patient group, 47 (55.29%) underwent lobectomy, and 38 (44.71%) had a total thyroidectomy. We recorded 16 cases (18.82%) with multifocal disease, 69 (81.18%) with lymph node metastases and 43 cases (43/85, 50.59%) of lateral cervical lymph node metastases in PM group. Postoperative treatment included thyroid-stimulating hormone inhibition and RAI therapies.

After the surgery, the patients were required to have follow-up visits every 3-6 months for the first year through cervical ultrasounds and thyroid functional examinations. The re-examination interval was then prolonged for patients with negative ultrasounds or CT, low serum thyroglobulin level, or no persistent disease. Disease remission and recurrence were determined according to the American Thyroid Association management guidelines^{11,12}. Disease remission was defined as two consecutive negative whole-body scans and ultrasounds. Due to inaccurate evaluation based on serum Tg and TgAb for patients with lobectomy, only structural recurrence was considered in this study. Disease recurrence was determined as a new disease in the thyroid bed or lymph nodes proven by cytology or histopathology, and/or confirmed by ultrasounds or CT scans, or distant metastases detected by whole-body scan. ”

Three clinical features are the risk factors of PPTC recurrence

Univariate Cox proportional hazard (CoxPH) model showed that age, TLNN, and LLNN may be risk factors for recurrence in pediatric patients. To determine the form of the age variable, the ten clinical features were next used as the inputs of multivariate CoxPH models.

Reply:

Thank you for your summary.

Protein qualification

Around 1272 proteins was removed with a missing value (NA) rate above 85%. CV of proteins were calculated cross the 16 pooled samples were mainly between 0.0 and 0.2, with a median of 0.0493 the CVs of the proteins across each pair of replicates were mostly less than 0.2, with medians of 0.0662, 0.0947, 0.1238, 0.0890, 0.0645 and 0.1123.

- Would recommend plotting correlation plot in which one half will be representing the intra sample correlation and other half inter sample correlation, thus one can determine the intra and inter sample correlation of individual sample.

Reply:

According to your suggestion, we plot the Pearson correlations for each paired sample. From the heatmap of correlation, we can find that the samples from the same group are well clustered, which indicates the substantial differences on proteotype. The low-risk PM samples are similar to the high-risk PMs based on the whole proteome.

Legend: Heatmap of Pearson correlation coefficients between each paired sample. Samples are labeled by the patient group. PM_Low and PM_high indicate the predicted risk of recurrence.

- One should mention the dimensionalities of data processing in terms of Normalization, Scaling and transformation

Reply:

In this study, we adopted a sequential strategy to preprocess our protein matrix. Firstly, to correct the plex-by-plex batch effects, we used internal reference scaling to convert the raw intensities into the relative abundances; then, after removing proteins with high missing proportions (> 85%), we imputed the missing values by *impseqrob* method, and finally, corrected the remaining batch effects by *ComBat* algorithm.

Proteomic differences among pediatric malignant, pediatric benign and adult malignant thyroid nodules

Around 243 was significant in (PM vs. PB) and 121 proteins were significant in (PM vs. AM) differentially expressed proteins (DEPs) with fold change (FC) > 1.5 and adjusted P < 0.05.

- Again the axis of Figure 3A, 3B and 3C are showing $-\log_{10}$ (Abundances of P value) vs $\log_2(\text{FC})$. Thus during the volcano plot (FC) or $\pm \log_2(\text{FC})$ was considered was not exclusive. One should mention this DEPs are both significant and differentially expressed as its fold change $(\text{FC}) \geq \pm 1.5$ and adjusted $P < 0.05$.

Reply:

Thanks for this suggestion. We have supplemented these DEPs derived from the FC 1.5 threshold in the caption of this figure.

"(C) The scatter diagram shows the FC distribution of the dysregulated proteins in two pairwise comparisons: PM vs. PB and PM vs. AM. Here, the DEP lists were derived from fold change threshold 1.5. "

- By lowering our FC threshold to 1.2, the number of DEPs increased to 1548 (PM vs. PB) and 1629 (PM vs. AM). What was the reason of lowering the FC, these are from FFPE tissue samples, and proteins must be significantly dysregulated as exceptionally observed for autoantibodies in serum/plasma samples.

Reply:

In this study, we used the tandem mass tag (TMT) labeling technique to quantify the proteome. The data we analyzed were adjusted by internal reference scaling firstly to mitigate the potential plex-to-plex batch effects, which means the protein abundances in our matrix were expression ratios rather than the intensities themselves. In each TMT 16-plex batch, there are 15 samples to be analyzed and one pooled sample. The relative protein abundance of 15 samples were represented by the ratios of the original intensities to the corresponding intensity of the pooled sample. The pooled samples are the same and act as a bridge between different batches. Therefore, compared to protein abundances measured using label-free quantification, these ratios have been compressed in terms of mean and scale. The fold changes derived from ratio expression are not as large as those from the original intensity, which is called Ratio Compression Effects¹. Thus, a lower threshold for fold change has been adopted in some published papers

using the TMT technique, such as \log_2 fold change 0.25 (fold change 1.189207)² and fold change 1.2³.

In the meantime, to avoid losing too many proteins by simple rules like FC, we adopted protein list from FC threshold 1.2 to give our model more freedom to decide by itself (though more time-consuming) which protein feature to use, which is a more precise way to build a more powerful model. In this way, the model would only select the most predictive protein combination. And the combinations derived from FC 1.2 threshold protein list would be much more and fully include those from FC 1.5 threshold, which leads to more (at least equally) powerful prediction model compared to that from FC 1.5 threshold.

- Authors have performed a differential proteome analysis between (PM vs. AM) and (PM vs. PB). To clarify the message for the scientific community, a functional clustering of differential expressed proteins between (PM vs. AM) and (PM vs. PB) is needed to be elaborated in main figure. This figure should complement the actual figure 3. Currently, this information is present in fragmented, volcano plots can be moved to figure S2A. The molecular functions their ORA or GSEA and their enrichment scores can be brought in Figure 3.

Reply:

Thank you for your comments. To reduce the fragmentation of information, we have placed all panels for the results of enrichment analysis and differential proteins (volcano plot) in the main Figure 3 in the updated version.

Immune infiltration and expression level of immune checkpoints in pediatric thyroid nodules

In-Silico tumour microenviron were studied using 'in-silico flow cytometry' CIBERSORTx. Seven types of immune cells were imputed namely CD8+ T cells ($P = 3.7 \times 10^{-12}$), macrophages ($P = 0.031$), dendritic cells ($P = 1.4 \times 10^{-5}$) and Treg cells ($P = 0.007$). CD8+ T cells and macrophages are increased in PM samples, while dendritic cells and Treg cells are reduced in PM

samples. CD8+ T cells and macrophages are increased in PM samples, while dendritic cells and Treg cells are reduced in PM samples which were validated using immunofluorescent staining.

Reply:

Thank you very much for the summary. We found that most regulated proteins in the comparison of PM vs PB are involved in the immune system. Therefore, we first performed CIBERSORT in-silico analysis to find the potential altered immune cells and then we validated the cells by immunofluorescent staining.

Analysis of 19 feature proteins

Random survival forest algorithm selected 19 proteins as features for the ProtRsf model, network analysis showed that 13 of the 19 protein features were directly or indirectly connected to disease.

- What about protein interactomes specifically modulated in each cohort comparison? Authors should complement their analysis including protein interactome networks, looking for specific hubs in each cohort comparison. To enrich the discussion section, are there specific transcription factors or upstream regulators that might be involved in the downstream modulation observed at proteome level in each cohort of PPTC comparison?

Reply:

To analyze the transcription factors or upstream regulators, we performed IPA with upstream module. There were four transcription regulators mapped with $P < 0.01$ for the 19-protein panel. The results are listed in the table below. We further searched the literature through PubMed and found that SREBF1 is a reported, prognostically relevant protein in thyroid cancer^{18,19}. The other proteins with $P < 0.01$, which have not been reported in studies related to thyroid cancer prognosis, are potential regulators that we identified. This part of the new results have been added to our revised manuscript.

Supplementary Table 7. Predicted transcription regulators of the 19 protein features

Molecule Type	Upstream Regulator	P value of overlap	Target Molecules in Dataset
transcription regulator	SREBF1	0.000977	GLDC, LGALS3, RNASET2
transcription regulator	ERCC6	0.00208	LGALS3
transcription regulator	CBFA2T2	0.00277	CHGA
transcription regulator	EOMES	0.00285	COL6A3, ITGA4

P values are calculated by Fisher's exact test.

- Authors can also perform targeted proteomic validation on selected protein features using SRM/PRM to see the trend of this proteins across each cohort of PPTC comparison.

Reply:

This is a good suggestion. In this case, we used TMT-labeled proteomics, the discovery proteomics strategy, which can get deep identifications to find the potentially altered pathways and functions. For further clinical translation, we are setting up a new multicenter prospective study to validate the protein panel by the cheaper and faster proteomic strategy, SRM. This part is ongoing and has not been included in the present study.

References

- 1 Savitski, M. M. *et al.* Measuring and managing ratio compression for accurate iTRAQ/TMT quantification. *J Proteome Res* **12**, 3586-3598, doi:10.1021/pr400098r (2013).
- 2 Shen, B. *et al.* Proteomic and Metabolomic Characterization of COVID-19 Patient Sera. *Cell* **182**, 59-72 e15, doi:10.1016/j.cell.2020.05.032 (2020).
- 3 Nie, X. *et al.* Multi-organ proteomic landscape of COVID-19 autopsies. *Cell* **184**, 775-791 e714, doi:10.1016/j.cell.2021.01.004 (2021).
- 4 Newman, A. M. *et al.* Determining cell type abundance and expression from bulk tissues with digital cytometry. *Nat Biotechnol* **37**, 773-782, doi:10.1038/s41587-019-0114-2 (2019).
- 5 Stewart, P. A. *et al.* Proteogenomic landscape of squamous cell lung cancer. *Nat Commun* **10**, 3578, doi:10.1038/s41467-019-11452-x (2019).
- 6 Wang, Y. *et al.* Longitudinal proteomic investigation of COVID-19 vaccination. *Protein Cell* **14**, 668-682, doi:10.1093/procel/pwad004 (2023).
- 7 Chen, Y. *et al.* Proteomic profiling of gastric cancer with peritoneal metastasis identifies a protein signature associated with immune microenvironment and patient outcome. *Gastric Cancer* **26**, 504-516, doi:10.1007/s10120-023-01379-0 (2023).
- 8 Newman, A. M. *et al.* Robust enumeration of cell subsets from tissue expression profiles. *Nat Methods* **12**, 453-457, doi:10.1038/nmeth.3337 (2015).
- 9 Iosef, C. *et al.* COVID-19 plasma proteome reveals novel temporal and cell-specific signatures for disease severity and high-precision disease management. *J Cell Mol Med* **27**, 141-157, doi:10.1111/jcmm.17622 (2023).
- 10 Iosef, C. *et al.* Plasma proteome of Long-COVID patients indicates HIF-mediated vasculo-proliferative disease with impact on brain and heart function. *J Transl Med* **21**, 377, doi:10.1186/s12967-023-04149-9 (2023).
- 11 Haugen, B. R. *et al.* 2015 American Thyroid Association Management Guidelines for Adult Patients with Thyroid Nodules and Differentiated Thyroid Cancer: The American Thyroid Association Guidelines Task Force on Thyroid Nodules and Differentiated Thyroid Cancer. *Thyroid* **26**, 1-133, doi:10.1089/thy.2015.0020 (2016).
- 12 Francis, G. L. *et al.* Management Guidelines for Children with Thyroid Nodules and Differentiated Thyroid Cancer. *Thyroid* **25**, 716-759, doi:10.1089/thy.2014.0460 (2015).
- 13 Memeh, K. *et al.* Total Thyroidectomy vs Thyroid Lobectomy for Localized Papillary Thyroid Cancer in Children: A Propensity-Matched Survival Analysis. *J Am Coll Surg* **233**, 39-49, doi:10.1016/j.jamcollsurg.2021.03.025 (2021).
- 14 Sudoko, C. K. *et al.* Thyroid Lobectomy for T1 Papillary Thyroid Carcinoma in Pediatric Patients. *JAMA Otolaryngol Head Neck Surg* **147**, 943-950, doi:10.1001/jamaoto.2021.2359 (2021).
- 15 Sugino, K. *et al.* Risk Stratification of Pediatric Patients with Differentiated Thyroid Cancer: Is Total Thyroidectomy Necessary for Patients at Any Risk? *Thyroid* **30**, 548-556, doi:10.1089/thy.2019.0231 (2020).
- 16 Baloch, Z. W. *et al.* Overview of the 2022 WHO Classification of Thyroid Neoplasms. *Endocr Pathol* **33**, 27-63, doi:10.1007/s12022-022-09707-3 (2022).
- 17 Jung, C. K., Bychkov, A. & Kakudo, K. Update from the 2022 World Health Organization

- Classification of Thyroid Tumors: A Standardized Diagnostic Approach. *Endocrinol Metab (Seoul)* **37**, 703-718, doi:10.3803/EnM.2022.1553 (2022).
- 18 Li, C. *et al.* SREBP1 as a potential biomarker predicts levothyroxine efficacy of differentiated thyroid cancer. *Biomed Pharmacother* **123**, 109791, doi:10.1016/j.biopha.2019.109791 (2020).
- 19 Kuo, C. Y. *et al.* SREBP1 promotes invasive phenotypes by upregulating CYR61/CTGF via the Hippo-YAP pathway. *Endocr Relat Cancer* **29**, 47-58, doi:10.1530/ERC-21-0256 (2021).

Reviewers' Comments:

Reviewer #1:

Remarks to the Author:

Authors have done an excellent job addressing the reviewers' comments. This reviewer and reviewer # 3 both mentioned the 1.2 FC vs 1.5 FC. The explanation that authors give to reviewers should be incorporated into the methods/results so readers are not confused. Some paraphrasing of this explanation would be acceptable.

Reviewer #2:

Remarks to the Author:

The author's response and edit's to the manuscript are satisfactory. I have no further questions. Thanks.

Point-by-point response to the reviewers' comments

REVIEWERS' COMMENTS (ROUND 2)

Reviewer #1 (Remarks to the Author):

Authors have done an excellent job addressing the reviewers' comments. This reviewer and reviewer # 3 both mentioned the 1.2 FC vs 1.5 FC. The explanation that authors give to reviewers should be incorporated into the methods/results so readers are not confused. Some paraphrasing of this explanation would be acceptable.

Reply:

We appreciate the comment. The reasons have been added in the Methods copied below.

“To avoid losing too many proteins by simple filters like FC, we adopted protein list from FC threshold 1.2 to give our model more freedom to decide by itself (though more time-consuming) which protein feature to use. In the enrichment analysis, to avoid over-complicated results without specificity caused by too many protein inputs, we used a strict threshold of FC 1.5. ”

Reviewer #2 (Remarks to the Author):

The author's response and edit's to the manuscript are satisfactory. I have no further questions.
Thanks.

Point-by-point response to the reviewers' comments

We thank all reviewers for their thorough, constructive and positive comments on our manuscript (NCOMMS-23-24388) entitled "*An individualized protein-based prognostic model to stratify pediatric patients with papillary thyroid carcinoma*". We believe that in the revised version, all points are now addressed, further strengthening our manuscript. In a nutshell, we included additional analyses, refined figures and text passages according to the suggestions. In the pages below, each of the reviewers' comments are addressed in more detail. We provide data directly in those cases where it was not appropriate to integrate into the revised manuscript.

Reviewer #1 (comments to the author):

Wang et al. have done a good job characterizing a large cohort of patients with papillary thyroid carcinoma. A major finding is development of a prognostic prediction model for pediatric papillary thyroid carcinoma. The results are important and could be publishable, but the reviewer has some concerns that need to be addressed first.

Reply:

We appreciate the positive summary of our manuscript. In the revised version, we provided the details of our data analysis, carefully checked the data and modified our manuscript according to the suggestions.

Line 29: 'Pediatric papillary thyroid carcinomas (PPTCs) are with high inter-tumor heterogeneity...' The wording here is a bit awkward. Consider rephrasing to something like "Pediatric papillary thyroid carcinomas (PPTCs) exhibit high inter-tumor heterogeneity..."

Reply:

Thank you for the suggestion. We agree it will flow better and more clearly conveys the intended meaning with rephrasing. This sentence has been revised in the manuscript as follows: "*Pediatric papillary thyroid carcinomas (PPTCs) exhibit high inter-tumor ...*".

Line 158: 1.2 fold change is a very, very low threshold. Authors need to justify its use here. Although these are statistically significant values based on adjusted p-value, how biologically relevant are they? Do the pathway enrichment (or modeling results) change significantly if a 1.5 FC threshold is used instead? IPA is mentioned elsewhere, but it is generally unclear what pathway enrichment software/database is being used to generate the results.

Reply:

We apologize for the confusion. In this study, we used the proteins derived from fold change (FC) threshold 1.5 for all the enrichment analyses and FC 1.2 for the feature selection in the process of model construction. The following are the detailed considerations when we chose the threshold 1.2.

1. Justification for why we used fold change threshold 1.2

In this study, we used the tandem mass tag (TMT) labeling technique to quantify the proteome. The data we analyzed were adjusted by internal reference scaling firstly to mitigate the potential plex-to-plex batch effects, which means the protein abundances in our matrix were expression ratios rather than the intensities themselves. In each TMT 16-plex batch, there are 15 samples to be analyzed and one pooled sample. The relative protein abundance of 15 samples were represented by the ratios of the original intensities to the corresponding intensity of the pooled sample. The pooled samples are the same and act as a bridge between different batches. Therefore, compared to protein abundances measured using label-free quantification, these ratios have been compressed in terms of mean and scale. The fold changes derived from ratio expression are not as large as those from the original intensity, which is called Ratio Compression Effects¹. Thus, a lower threshold for fold change has been adopted in some published papers using TMT technique, such as log₂ fold change 0.25 (fold change 1.189207)² and fold change 1.2³.

2. Fold change choice in modeling

In the meantime, to avoid losing too many proteins by simple filters like FC, we adopted protein list from FC threshold 1.2 to give our model more freedom to decide by itself (though more time-consuming) which protein feature to use, which is a more precise way to build a more powerful model. In this way, the model would only select the most predictive protein combination. And the combinations derived from FC 1.2 threshold protein list would be much more and fully include those from FC 1.5 threshold, which leads to more (at least equally) powerful prediction model compared to that from FC 1.5 threshold.

3. Fold change choice in pathway enrichment analysis

In our manuscript, the IPA pathway and GO enrichment analyses were based on the protein list derived from fold change threshold 1.5 (including 243 proteins in PM vs. PB and 121 proteins in PM vs. AM). We used FC 1.5 as our threshold rather than 1.2 because there are too many proteins (more than 1000) in the protein list derived from FC threshold 1.2 and thus too many pathways (or biological processes, molecular functions, cellular components) would be enriched with high significance, leading to over-complicated results without specificity. And the enrichment results from 1.2 and 1.5 FC proteins are the same for the most significant changes, which remain to be related to immune system dysfunction.

Finally, we supplemented the enrichment methods we used for each panel in both figure legends and methods sections.

"Pathways and networks were analyzed using the Ingenuity Pathway Analysis (IPA) and visualized with Cytoscape (v3.8.2). GO enrichment analysis was conducted by enrichGO function in R Package clusterProfiler (v4.0.5) using database org.Hs.eg.db (v3.13.0, stored in R package org.Hs.eg.db)."

Line 163: 'According to the enrichment analysis of annotated keywords performed using STRING database, the most upregulated proteins in PM, compared to the other two groups, were involved in MHC-II and immunity.' Please provide the enrichment analysis results in the manuscript (or in supplemental). Is it surprising that all of the top pathway results are immune related? Were there other pathway results with better p-values that were not shown? Are these pathway hits p-values or adjusted p-values? Are these still significant when adjusted p-values are considered?

Reply:

Thank you for the suggestions. We have provided the details in the supplemental files. Results from the STRING database are in the **Supplementary Table 4**. These immune-related results shown in our manuscript are all based on significantly dysregulated pathways or function enrichments. In order to confirm the reliability of these results, we used a variety of enrichment software or methods, *i.e.*, IPA, clusterProfiler, STRING, METASCAPE, and all of which pointed to immune system-related, a result that was indeed very surprising to us as well. This is why we chose to further increase the analysis of immune cell infiltration in **Figure 4**.

All pathways we present were not artificially selected or filtered. We used *P* value rather than adjusted *P* values in our manuscript. According to your suggestion, we also tried BH-adjusted *P* values as the cutoff; the pathways we presented still remained.

Line 168: What proteins are used for the enrichment? The 1.5 fold change or the 1.2 fold change? Please state in the text. Pathway enrichment results should be provided in supplemental materials.

Reply:

We used a 1.5-fold change as the cutoff. The two DEPs lists are added in **Supplementary Table 3** and detailed enriched pathways and functions are listed in **Supplementary Table 5**.

Line 179: Authors should justify use of a gene deconvolution tool CIBERSORTx on proteomics data and provide evidence that this is working as expected. Immune cell deconvolution based on

gene signatures requires dozens if not hundreds of genes. How good is the coverage of the genes in gene signatures given this is proteomics data? Are these Welch's t-test p-values? Is the condition of normality satisfied for using a t-test here? Do the results/significant differences change if a non-parametric test like Wilcoxon rank-sum test is used?

Reply:

Since multiple dysregulated immune-related pathways and biological processes were enriched, we would like to further explore the difference of immune microenvironment in PB and PM samples. Thus, we conducted the immune infiltration analysis using CIBERSORTx. Our customized gene signature matrix contains 2223 genes, which is derived from the published RNAseq data of thyroid papillary carcinoma. Among the 2224 gene features, 1451 (65.3%) genes can be matched to the protein abundances, which is higher than the 50% threshold suggested by the software. The authors of CIBERSORTx have proposed possible estimation biases introduced by inter-platform variations. They have provided a batch correction method as an optional parameter in CIBERSORTx to deal with this problem, and the deconvolution results substantially improved compared to the ground truth⁴. We applied the batch correction provided in CIBERSORTx to our analysis to reduce the influence of inter-platform bias. Also, CIBERSORTx has been used in many proteomic studies⁵⁻⁷.

In our submitted manuscript, we used Welch's *t*-test, which was done by the `stat_compare_means` function in R package *ggpubr*. We tested the data for normality using Shapiro-Wilk's test and found that out of the 14 sets of data (7 cell types * 2 datasets), 6 sets of data did not fit the normal distribution and 8 sets did. If the Wilcoxon Rank-Sum Test is used instead, the results are shown below. $P < 0.05$ for all cell types except CD4+ T cells, but only CD8+ T cells and dendritic cells meet $P < 0.01$.

The statistical analysis method has been added in the revised legend of **Figure 4**.

Legend: Relative proportions of seven types of immune cells in PB and PM samples imputed by CIBERSORTx. The significance was determined by Wilcoxon Rank-Sum Test. (Not used in our manuscript)

Line 222: Given the prominence of immune-related findings earlier, is it surprising that there were not more immune related hits here? Were any of these proteins also differentially expressed? Do these proteins belong to related pathways or families?

Reply:

We think that is not surprising. These two lists of proteins represent two aspects. The dysregulated proteins related to immune systems are from the differences between benign and malignant pediatric thyroid nodules. However, proteins mentioned in Line 222 evolved from the differences between pediatric papillary thyroid cancers with recurrence and un-recurrence. To avoid data leakage, we didn't use the differentially expressed proteins (DEPs) between recurrence and un-recurrence groups to build the predicting model. Instead, we applied the DEPs from malignant vs. benign groups with 1.2 FC as the protein candidate.

We did the network analysis for the selected protein in Line 222. Our network analysis showed that 13 of the 19 protein features were directly or indirectly connected. In particular, LGALS3, the hub protein, may perform a significant role in pediatric thyroid carcinoma (**Figure 5E**).

Line 237: 'ITGA4 and GAL3ST4 were found positively correlated to CD8+ T cells in both groups.' Whenever a quantitative statement is made, the value and corresponding p-value should be provided. Please fix here and throughout.

Reply:

Thank you for your suggestion. The modified text reads as follows.

*"ITGA4 ($P = 7.28 \times 10^{-4}$ and $P = 2.05 \times 10^{-8}$ in high and low risk groups, respectively) and GAL3ST4 ($P = 1.56 \times 10^{-4}$ and $P = 1.24 \times 10^{-4}$ in high and low risk groups, respectively) were found positively correlated to CD8+ T cells in both groups. For the 31 immune checkpoint proteins quantified, only the abundance of IL10RB was found to decrease with the predicted recurrence risk and highest in PB samples ($P = 0.0012$ and 0.038 , respectively; **Supplementary Figure 3**)."*

Line 231: Wilcoxon did not seem to be mentioned in the methods. Are these Wilcoxon rank-sum tests, specifically? Please be more specific.

Reply:

Thank you for your kind reminder. For **Figure 5F**, two-sample Wilcoxon Rank-Sum Tests were used to estimate the P values. We refined statistical tests for each figure and added the names of tests legends.

Line 242: Are there any independent cohorts with protein expression (or even gene expression) for validating the signature? Translational impact of the manuscript could be much higher if the signature can be validated with independent samples (or experiments). How well does the signature work in adults? Could TCGA gene expression data from (adult?) thyroid cancers be used as some form of validation?

Reply:

We recognize that additional center and sample validation can further confirm the reliability of the proteins selected by this method. However, this study is currently a single-center cohort study, which is a limitation of this study at this time. In the future, we hope to further confirm the robustness of our biomarkers through prospective multicenter clinical trials.

There are significant differences in both protein expression and phenotype between thyroid cancers in children and adolescents and those in adults, so it is reasonable to assume that this method is not applicable to adults. Recurrent papillary thyroid cancer in adults is another project in our team. In addition, according to previous studies, it has been demonstrated that the expression at the mRNA level correlates less than 60% with the expression at the protein level, and therefore the protein information is not directly transposable to the mRNA molecular level. Therefore, if it is necessary to analyze from the mRNA level, it is necessary to construct a separate classifier. We investigated the TCGA data, and there were only 14 cases of thyroid cancer in pediatric (all of them were older than 14 years old). More importantly, there was no recurrence information provided. Thus, TCGA data is not possible to be used for validation.

Line 419: More PD parameters are required to enable replication of results. Did authors use TMT channel bleed-over correction setting in PD?

Reply:

Thank you for your advice. We added the detailed information in the methods section, as shown below.

"Proteomic raw files were searched using Proteome Discoverer (v2.4.1.15) against a FASTA file containing 20,368 entries (human Swiss-Prot database). Channel TMT-126 was set as the reference for each batch. Correction factors (Lot# VG306794) were used when we did the data searching. The search parameters were set as follows: two trypsin missed cleavages allowed; minimal peptide length of 6 amino acid residues; precursor ion mass tolerance of 10 ppm; fragment mass tolerance of 0.02 Da. Normalization was performed against the total peptide amount. The false discovery rate thresholds were set to strict 1% for peptide and protein quantification. Other settings

were left to their default values."

Line 430: Did authors follow internal reference scaling (IRS) to remove plex-to-plex batch effects? If not, then how were the pools used? Anecdotally, IRS seems to outperform ComBat for removing plex-to-plex batch effects. Authors should consider trying this method to see if it improves differential expression results/model prediction.

Reply:

Yes, we have incorporated internal reference scaling (IRS) in our analysis. Immediately after the database search, IRS was used to correct the plex-to-plex batch effects in the protein expression matrix. Specifically, we added a pooled sample in each batch, and the relative protein abundance of the other 15 samples in the same batch were calculated by dividing the intensities of the pooled sample. The pooled samples are the same among all batches, so the pooled samples acted as internal references here.

However, after quality control analysis and missing value imputation, batch effects were still detected by UMAP as shown in the below left figure. Thus, to further correct the batch effects, we applied ComBat for removing the remaining batch effects. And after sequential batch effects correction (IRS + ComBat), the batch effects were no longer significant, as shown in the below right figure or **Supplementary Figure 1E**.

UMAP show the batches of sample distribution before (left) and after (right) batch correction using ComBat.

Line 434: Welch's t-test is used line but then Student's t-test is mentioned line 445-446. Why not use Welch's t-test throughout? Why assume equal variances here for these checkpoint proteins?

Reply:

We apologize for the mistake in line 445-446. It was actually a Welch's *t*-test (the default option for *t*-test) conducted using 'stat_compare_means' function of R package *ggpubr*. Thank you again for pointing this out.

Line 431: Although the ComBat approach does result in negative abundances, the reviewer thought this was okay because the transformed values are changed into some other arbitrary units. If these are being treated as relative abundances, then the sign of the value might not matter as much as the differences in abundances across groups. If having negative values is okay, then arbitrarily replacing negative values can alter the distribution of the transformed data and adversely impact the downstream analyses. Please provide additional justification for handling the negative values or please change how the data is processed.

Reply:

Thanks for this suggestion. After internal reference scaling, the abundances in the protein matrix become the expression ratio. And these relative abundances are relative to the pooled samples, which represents how many folds of the abundances in analyzed samples when compared with the abundances in pooled samples. In this case, negative values are meaningless because we cannot expect negative folds relative to the abundance of pooled samples. Another reason is that if we did not deal with these negative values, then for some proteins, their sums across groups (*e.g.*, PB, PM, AM) would be negative, resulting in meaningless negative fold changes when doing downstream analyses. Therefore, we replaced these negative values with half the minimum

value of the positive abundances of the corresponding protein to maximumly keep the differences in abundances across groups.

Lines 433-434: The differentially expressed proteins (DEPs) were identified with fold change (FC) values to be greater than 1.2 or 1.5 (for different purposes)' What different purposes? Please explicitly state here in methods so readers do not need to search through all of the results.

Reply:

According to your suggestion, we clarify these in the method section.

"The differentially expressed proteins (DEPs) were identified with fold change (FC) values to be greater than 1.2 or 1.5 (1.2 for modeling and 1.5 for enrichment analysis), with an BH-adjusted Welch's t-test $P < 0.05$."

Lines 441-442: 'We used a custom signature matrix generated from published thyroid cancer single-cell RNA data'. Given the disconnect between gene expression and protein expression (and even the disconnect between single cell gene expression and bulk gene expression), please justify the use of this custom signature matrix. Has this been previously validated in other proteomic datasets?

Reply:

Thanks for the comment. The question is also explained in the previous comments.

To minimize the impact of cross-platform variation on the deconvolution results, there are batch-correction options provided in CIBERSORTx. The method has been demonstrated to be useful for cross-platform data⁸, for example, deconvolute cell fractions from bulk RNA-seq data using a scRNA-seq signature matrix. For proteomic data, the microarray-derived leukocyte gene signature matrix provided by CIBERSORTx, termed LM22, has been used to infer immune cell infiltration⁶. LM22 contains 547 genes that distinguish 22 human hematopoietic cell phenotypes. The reason we do not use the built-in LM22 matrix is that it is derived from peripheral blood cells and only 175/547 were identified in our matrix. Hence, we customized the signature matrix using

published scRNAseq data of papillary thyroid tumors and performed the S-mode correction which is tailored for single cell-derived signature matrices when deconvolute our data to correct the cross-platform bias. By this way, we quantified 65.3% (1451 proteins out of 2223 genes) features in thyroid thyroid-specific matrix. As for using gene expression signature matrix to deconvolute proteomics data, similar applications can be seen in many other studies^{6,7,9,10}

Line 481: Information about feature selection for the modeling should be briefly included in results too.

Reply:

Thank you for your advice. We have supplemented the related content in the Results section.

"To predict the PTC recurrence risk of patients from the PM group, the PM samples were randomly divided into a training set (n=50, ~60%) and an independent test set (n=35, ~40%). Then, we developed five models based on two algorithms (Cox proportional hazard model and random survival forest) and two types of features (clinical features and proteins). Specifically, we developed the following models: two Cox proportional hazard models based on clinical features (CliCox) or protein features (ProtCox); three random survival forests based on clinical features (CliRsf), protein features (ProtRsf), or clinical and protein features (CliProtRsf). For each model, we tuned the hyperparameters using grid search strategy and 3-fold cross-validation, selected the features, and trained the model using the training set. The final hyperparameter settings of the five models are summarized in Supplementary Table 2."

Line 517: Identifier IPX0006407000 did not return any results when searched on iProX. The reviewer assumed <https://www.iprox.cn/> is the correct website, but more information and possibly a doi/url should be provided. Although it is deposited in iProX, will it be indexed by the ProteomeXchange Consortium? Please make sure data is available.

Reply:

Yes, iProX is an integrated proteome resources center in China and is indexed by the ProteomeXchang Consortium. Please visit our data through URL:

<https://www.iprox.cn/page/DSV021.html?url=1705059006234WK3k>; Password: yhkB. All the data will be open to the public once the paper is published.

Authors should provide all protein abundance matrices, pathway enrichment results, etc. as part of supplemental materials.

Reply:

We have uploaded the detailed patient information table and proteome matrix (**Supplementary Table 2**), DEPs lists (**Supplementary Table 3**), and enrichment analysis results (**Supplementary Table 4 and 5**) as supplemental files.

Authors filled out a form with a link to a github repository that was empty. Please make sure code is available that can be used to reproduce the modeling results. A link to the working github repository should be included in the manuscript.

Reply:

Thanks for pointing out this. We have uploaded our codes for analysis. The repository link is <https://github.com/wanghe98/PPTC>

Line 722: IHC images are very small. Please make them bigger. Does the 50um scale bar apply to all images? The scale bar should be included in all images or this should be directly stated in the text.

Reply:

Thank you for your suggestions. The figure has been modified, and the scale bar is stated in the figure legend.

Reviewers #2 (comments to the author):

Wang et al's manuscript is an interesting and novel approach to create a clinical path to predicting the risk of recurrence in pediatric patients diagnosed with papillary thyroid cancer. As the authors accurately state, a 'one-size-fits-all' treatment strategy is associated with the potential for over-treatment of patients with low-invasive disease and, potentially, under-treatment of patients with invasive disease.

The current commercially available, molecular diagnostic panels used to augment cytological data are based on detection of somatic oncogenic driver alterations. Unfortunately, within each oncogenic driver, there is significant variability in the extent of invasive disease and response to therapy where additional, multi-omic data, including proteomic data, may provide further insight in predicting clinical behavior of the tumor.

Wang et al's proposed manuscript provides a first look at how proteomics may be used to predict the risk of recurrent disease in pediatric patients with PTC. The established expertise of the research group, depth of proteomic analysis, the analysis of five different models for data analysis, and assessment of the 19-protein model panel to analyze recurrent risk for high- versus low-risk patients. The stated limitation of the study is the lack of investigation of fine needle aspiration samples with prospective clinical follow-up to validate the reliability and accuracy of the proposed 19-protein panel model.

Reply:

We deeply appreciate the positive comments on the novel approach of our study and the acknowledgment of strengths such as our research expertise, depth of proteomic analysis, multiple modeling approaches, and focus on pediatric papillary thyroid cancer.

In response to the critique on the lack of validation with clinical follow-up, we fully agree that prospective confirmation is needed before clinical implementation. We have modified the Discussion section to note validation in an independent dataset as an important next step for future studies.

Thank you again for the thoughtful feedback, which has helped improve our work.

Questions

1) The authors report that patients were followed with cervical neck ultrasound and thyroid labs every 3 to 6 months after initial surgery, with increasing time between surveillance if patients had no evidence of persistent disease (Methods, line 355). The authors define remission as two consecutive, negative WBS and ultrasound with Tg and TgAb in the 'ideal range' (line 357-359). The authors define recurrence as structural or biochemical, with the latter defined by unstimulated Tg > 1 ng/mL, stimulated Tg > 10 ng/mL or increasing TgAb levels (line 359-364). These definitions are critical to analyzing the data as they form the basis of building and interpreting the proteomic data and model.

Of the 85 pediatric patients with PTC, the authors report that 47 (55.29%) underwent lobectomy and 38 (44.72%) underwent total thyroidectomy (Results, line 95). All patients underwent prophylactic central neck dissection (Results, line 96) and patients that underwent total thyroidectomy received RAI therapy (Methods line 353).

☐ The text currently states the definition of biochemical remission of Tg and TgAb as levels in the "ideal range". Is the "ideal range" for Tg and TgAb used to define remission the same as the for recurrence?

Reply:

We apologize for the unclear description in our manuscript.

We defined these two states according to the criteria in 2015 ATA guidelines^{11,12} copied below. Considering Tg and TgAb is not accurate for assessing recurrence in patients who underwent lobectomy, we did not apply the criteria for biochemical recurrence in the present study since some of the patients underwent lobectomy. In other words, all recurrent patients in our study are structural recurrences.

To make it clear, we modified the Methods section (deleted the biochemical recurrence).

"Disease remission and recurrence were determined according to the American Thyroid Association management guidelines^{11,12}. Disease remission was defined as two consecutive negative whole-body scans and ultrasounds."

"Due to inaccurate evaluation based on serum Tg and TgAb for patients with lobectomy, only structural recurrence was considered in this study. Disease recurrence was determined as a new disease in the thyroid bed or lymph nodes proven by cytology or histopathology, and/or confirmed by ultrasounds or CT scans, or distant metastases detected by whole-body scan."

☒ Can the authors explain how remission and recurrence was defined for patients that underwent lobectomy? The Tg and TgAb definitions provided in lines 359-364 would be consistent for patients that underwent total thyroidectomy + RAI but cannot apply to patients that underwent lobectomy.

Reply:

Patients who underwent lobectomy were also only considered for structural recurrence. The definition of recurrence for patients with different surgical procedures has been modified in the manuscript and stated above.

☒ For the patients that had recurrent disease (n = 10, Results, line 245), how many had undergone lobectomy versus total thyroidectomy + RAI?

Reply:

In our study, there are 12 patients had recurrence. Six patients underwent lobectomy, and six patients underwent total thyroidectomy, with three of them undergoing total thyroidectomy + RAI.

The ten patients mentioned in line 245 refer to the number of patients that were wrongly identified by our model, comprising two false negatives (lobectomy n=1 versus total thyroidectomy n=1) and eight false positive cases (lobectomy n=7 versus total thyroidectomy n=1).

☐ How did the authors incorporate the differences in reliability and accuracy of remission for lobectomy vs total thyroidectomy + RAI into the model predicting recurrent disease?

Reply:

This is a good question. Currently, it is controversial regarding the effect of the surgical procedure on the recurrence of pediatric PTC¹³⁻¹⁵. In our cohort, we found no significant difference in the effect of surgical approach on recurrence.

Recurrence was observed in 12.77% (6/47) and 18.42% (7/38) of patients undergoing lobectomy and total thyroidectomy, respectively. To explore the effect of the surgical approach on recurrence, we performed chi-square test, Kaplan-Meier survival analysis and Cox regression model. The chi-square test showed no significant difference ($\chi^2=0.158$, $P=0.691$), the Kaplan-Meier survival curves suggested that the difference in recurrence-free survival (RFS) between the two surgical approaches was not statistically significant ($P=0.637$), and the univariate analysis of the Cox hazard proportional regression model showed that the surgical approach was not a risk factor for recurrence in patients ($P=0.638$, HR=0.762, 95% CI:0.245-2.365). The results of multivariate CoxPH models also indicate no substantial differences. Therefore, we did not include this factor in the model to predict recurrent disease in subsequent studies.

We added the new results and revised **Figure 2B** and **2C** as shown below.

Figure 2. Analysis of the clinical recurrence risk factors for PPTC. (B-C) Forest plots for two multivariate CoxPH models using (B) continuous non-negative integer age and (C) categorical age, respectively. *P* values are tested by Cox proportional hazard model.

Is there any differences, improvement or decrement, in the predictive accuracy of the model if the authors only include patients that received total thyroidectomy and RAI?

Reply:

The previous analysis indicated that the surgical approach was not a risk factor for prognosis in our cohort, and there was no difference in the proportion of the number of recurrences in each group, so total thyroidectomy patients were not studied separately. Secondly, the number of patients who underwent total thyroidectomy was too small (n=38). Studying only patients received total thyroidectomy halved the study population in the experimental group and the

number of patients with observed recurrence. The decrease in cohort size was not conducive to the modeling and subsequent validation of the results. Therefore, we chose to include all pediatric patients with thyroid cancer who had either lobectomies and total thyroidectomies.

2) In previous studies, both the PTC variant/subtype as well as the somatic oncogenic alteration have been shown to correlate with invasive behavior, including the risk for regional as well as distant metastasis. Patients that present with unifocal, encapsulated thyroid cancer, associated with low-invasive oncogenes (PTEN, DICER1, RAS) have a low risk for metastasis and a high likelihood of stable remission. In contrast, patients presenting with tumors having invasive/lobulated margins and high-risk oncogenes (i.e., BRAF V600E and fusions; PMID 35015563) have an increased rate for both regional and distant metastasis with the most common, associated PTC variants being classic PTC, diffuse sclerosing variant PTC, and widely invasive follicular variant PTC.

The authors report exclusion of PTC' subtypes with highly invasive disease, i.e., tall-cell variant, columnar and PDTC' (Methods, line 344) and do not provide any somatic oncogene data.

However, the cohort has a nice variance in regard to the breadth of regional metastasis, with 82% having LN metastasis, 31% (n = 26) with N1a and 51% (n = 43) with N1b disease. Only 1 patient with M1 disease is included (Supplement table 1).

□ What PTC subtypes are included in the study? How many with cPTC? Diffuse sclerosing PTC, and widely-invasive follicular variant PTC?

Reply:

Thank you for your question. To accurately answer this question, two additional histopathologists helped to carefully revisit HE slides for pediatric PTC. If the area of the particular pathological subtype accounted for more than 30% or more of the entire section, it was determined to be the corresponding pathologic subtype¹⁶.

From the 85 patients, there are 69 cPTC (81.2%), eight diffuse sclerosing variant PTC (9.4%), six hobnail variant PTC (7.1%), one solid variant PTC (1.1%) and one columnar cell variant PTC

(1.1%). For the twelve patients with recurrence, ten of them are cPTC, one diffuse sclerosing variant PTC and one hobnail variant PTC. With the PTC subtype information, we have modified the corresponding contents in Methods section.

☒ What is the somatic genotype for the PTC tumors included in the analysis and did the oncogenic driver correlate with the model predicting persistent/recurrent disease? i.e. is there a clinical advantage to using the proposed 19-gene model over currently available oncogenic driver panels?

Reply:

We do not have information about the genetic changes in these samples for the time being. In China, genetic testing is not a routine detection, and only a small number of patients choose to be tested for the *BRAF* V600E mutation. Among these 85 samples of pediatric PTC, nine patients were tested for *BRAF*, 5 of them had *BRAF* mutations. We believe that the integration of protein information with gene information will further improve the efficacy of the model, and this is one of our future planned work.

☒ If proteomics is brought into clinical practice, what is the estimate cost difference between a comprehensive, somatic oncogene panel vs. the 19-protein model panel from this study?

Reply:

Thank you for the comments regarding the practicality of using this assay in the real world. Our study is a proof-of-principle to show that protein-based classifiers can be used to classify the recurrence risks for pediatric PTCs. Right now, we can tell that the cost to measure 19 proteins using targeted proteomics is not higher than that for measuring 19 mRNAs. However, we respectfully maintain that this is beyond the scope of this manuscript.

3)Minor

☒ Line 64 -> reference #5 is not applicable as this reference examines radiation induced PTC (excluded from the study cohort)

☒ Line 70 -> references 8-10 are very outdated and BRAF is reported in about 40% of pediatric PTC. Consider replacing with reference #6 as well as PMID 35015563).

Reply:

Thank you for pointing out these issues and recommending proper literature. We have corrected them in the revised version.

4)As the authors suggest, a multicenter, prospective study in pediatric patients using FNA is needed to validate the 18-protein panel. It would also be interesting to test the panel's ability to predict patients that present with M1 disease.

Reply:

We appreciate the reviewer's suggestions regarding validation and testing the protein panel's ability to predict metastases. As the samples with distant metastasis (M1) were limited in our current cohort (only one case), the prediction of metastatic disease represents a separate issue that requires investigation in larger patient groups in future studies.

5)There is increasing data on how to stratify surgery and surveillance based on the sonographic features of thyroid tumors and cervical neck lymph nodes with somatic oncogene driver and gene sequencing. The burden on the authors is to show that this 18-protein panel provides a more reliable and accurate tool in identifying patients at risk for recurrent disease above the current standard of care. If the panel is validated for FNA, do the authors envision this as an additional, complimentary tool to somatic oncogene analysis?

Reply:

Thank you for your comments. We believe that protein panel can be a useful tool as an additional approach in the diagnosis and evaluation of thyroid diseases. Two keywords are worth being emphasized here, one is FNA, and the other is multimodal or multilevel information. FNA is the

gold standard for preoperative diagnosis, and if it is possible to detect both genes and proteins in one FNA-biopsy sample, it can further improve the assessment of cytopathology and, at the same time, provide more information on alterations at the molecular level for risk stratification of tumors, thus guiding a more appropriate treatment. We plan to initiate larger multicenter, prospective studies in subsequent studies and validate them by means of more accurate targeted proteomics.

The second point is that multimodal data has been a hot word in recent years. Multi-dimensional information can depict the state of the tumor more comprehensively and view the tumor from different perspectives, thus obtaining a more accurate assessment, which of course cannot be separated from the support of big data and artificial intelligence. This is also one of our future endeavors.

Reviewer #3 (Remarks to the Author): expertise in proteomics of endocrine cancer

In this work, Wang et al. performed detailed individualized protein-based prognostic model to study papillary thyroid in pediatric benign, pediatric malignant and adult malignant individual. Wang et al. study used machine learning-based based algorithm to predict classifier contribution to tumour recurrence and simultaneously stratified PPTC patients into high and low recurrence risk group.

The experimental design is good and the statistical control in all phases is appropriate. However, I consider that some major & minor points along the manuscript should be addressed:

Comments

The manuscript sections should be reorganized to facilitate the understanding and reading

Reply:

Thank you for your positive feedback on our paper. We highly appreciate your comments. This helps us improve the quality of the paper and better serve the academic community and clinical practice with the research findings. The following are our point-to-point replies.

Study population and samples collection

a. The clinical samples were collected between November 2007 and April 2021 for around 15.5 years. The IEC / IBSC Study number with First Hospital of China Medical University is missing.

Reply:

Based on your comments, we made additions here with the following revisions.

"This study was approved by the Ethics Committee of the First Hospital of China Medical University with the study number 2021-287-2."

b. Line No. 356 After the surgery, the patients were followed up every 3-6 months with cervical ultrasounds and thyroid functional examinations. In an average duration of 15.5 years how many times were these patients followed with a magnitude of (follow-up of 3-6 months).

Reply:

Thank you for the questions. In this study, all the postoperative patients visited the hospital every 3-6 months for the first year after surgery. Re-examination was prolonged for patients with negative ultrasounds or CT, low serum thyroglobulin level, or no persistent disease. Patients are asked to be followed up 6-12 months in the second to fifth year after surgery and every 1-2 years after five years of surgery.

c. Line no. 357- Re-examination was prolonged for patients with negative ultrasounds or CT, low serum thyroglobulin level, or no persistent disease (How many patients).

Reply:

According to our statistics, 50 patients were followed up for more than five years, 33 patients were followed up for at least one year, and only two patients followed up for less than one year.

Follow-up (years)	Total patients	Recurrent patients	Recurrent-free patients
≤1	2	1	1
1-5	33	8	25
>5	50	3	47

d. Line360- 3 Two types of recurrence, Structural recurrence based on imaging techniques and Biochemical recurrence based on clinical parameters, were studied for Cox proportional hazard (CoxPH) model with combined prognosis information: recurrence events, the time interval between surgery and recurrence, or between surgery and the last follow-up.

Each Clinical features identified the factors whose P values were less than 0.05, were taken forward. In a holistic view how many features (10 clinical features) were significantly, uniformly had a probability range of 0 to 1 in a combined analysis.

Reply:

Among the clinical features in univariate CoxPH model, three (Age, LNN, LLNN) were significant. But in the multivariate CoxPH model, only age (treated as a binary variable) was nearly significant.

e. There should be a paragraph titled 'Experimental Design and Statistical Rationale' that describes the number of biological and technical replicates analyzed and justification for how this provides statistical significance for the results reported. The information is scattered and need to be put together

Reply:

In response to your suggestion, we have rearranged and added the following paragraph to the methods section.

"Experimental design and statistical rationale

We collected FFPE slides for proteomics data acquisition. Each slide was stained with hematoxylin and eosin and reviewed by at least two experienced histopathologists and histopathological subtypes for PM were further evaluated according to The 15th edition of the World Health Organization Classification of Endocrine and Neuroendocrine Tumors¹⁷. Each slide was reviewed and processed to make sure the tumor ratio was approximately more than 80% before proceeding to proteomic sample preparation.

We collected 240 thyroid nodules (87 PM, 85 PB, 68 AM) FFPE slides (10 μ m thickness) from 234 patients (85 PM, 83 PB, 66 AM). Two samples from each group were randomly selected as technical replicates. To minimize the potential artificial effects during experiments, we randomly allocated the 240 tissues into 16 batches. In each batch, there were 15 tissue samples and one pooled sample was used as an internal reference scaling for the batches. The replicates and pooled samples were analyzed for data quality control."

Proteomics data acquisition and data analysis

Around 240 thyroid nodules were collected from 234 patients, the FFPE samples were processed by the protocol optimized by Zhu Y et al and Nie X et. al a, cleaned peptides were labelled with TMTpro 16-plex reagents and analysed in DIONEX UltiMate 3000 RSLCnano System and Orbitrap Exploris 480 with FAIMS Pro™. The data were analysed using Proteome Discoverer (v2.4.1.15). Tumour microenvironment was then studied with CIBERSORTx, which led to the identification of 7 immune cell types which is further analysed for immune check points. mIHC Staining of FFPE tumour sections were also performed for mainly CD3, CD4 and CD8.

a. However during the Proteome Discoverer analysis, the ability to account for reporter ion isotopic impurities for TMT10plex reagents (correction factors while designing the study); implementation of TMT quantification based on S/N values was not mention

Reply:

Thank you for the advice. Correction factors (Lot# VG306794) were used when we did the data searching. The S/N value was 1.5, which is the default setting. This information is added in the revised methods section.

b. Have the authors also accounted for the TMT labelling efficiency

Reply:

Yes, we have checked the TMT labeling efficiency and then combined the labeled peptides..

c. Would like to know whether its FC or Log2FC, FC difference of 1.2 is very less.

Reply:

We apologize for the confusion. In this study, we used the proteins derived from fold change (FC) threshold 1.5 for all the enrichment analyses and FC 1.2 for the feature selection in the process of

model construction. The following are the detailed considerations when we chose the threshold 1.2.

1. Justification for why we used fold change threshold 1.2

In this study, we used the tandem mass tag (TMT) labeling technique to quantify the proteome. The data we analyzed were adjusted by internal reference scaling firstly to mitigate the potential plex-to-plex batch effects, which means the protein abundances in our matrix were expression ratios rather than the intensities themselves. In each TMT 16-plex batch, there are 15 samples to be analyzed and one pooled sample. The relative protein abundance of 15 samples were represented by the ratios of the original intensities to the corresponding intensity of the pooled sample. The pooled samples are the same and act as a bridge between different batches. Therefore, compared to protein abundances measured using label-free quantification, these ratios have been compressed in terms of mean and scale. The fold changes derived from ratio expression are not as large as those from the original intensity, which is called Ratio Compression Effects¹. Thus, a lower threshold for fold change has been adopted in some published papers using TMT technique, such as log₂ fold change 0.25 (fold change 1.189207)² and fold change 1.2³.

2. Fold change choice in modeling

In the meantime, to avoid losing too many proteins by simple filters like FC, we adopted protein list from FC threshold 1.2 to give our model more freedom to decide by itself (though more time-consuming) which protein feature to use, which is a more precise way to build a more powerful model. In this way, the model would only select the most predictive protein combination. And the combinations derived from FC 1.2 threshold protein list would be much more and fully include those from FC 1.5 threshold, which leads to more (at least equally) powerful prediction model compared to that from FC 1.5 threshold.

d. How much peptide was injected in column

Reply:

For each batch, there were 112 ug of peptides (7 ug/channel*16 channels). The peptides were further separated into 120 fractions with 120 HPLC gradient and then combined into 30 fractions (#1,#31,#61,#91 into one, and so on). For each combined fraction, we first dried them and then resolved them using 30 uL MS bufferA, and inject 4 uL peptides into HPLC-MS/MS.

Predicting PPTC recurrence risk, stratification

PPTC was predicted with training data set of 60% and independent test data set of 40% using Cox proportional hazard model, random survival forest on both clinical and protein features. The training data set was utilized in hyperparameter tuning, feature selection, and model training and independent test set was used to evaluate our models. 3-fold cross-validation, selected the features, and trained the model using the training set. In the ProtCox model LASSO was used for selecting the protein features. 1,548 DEPs (PB versus PM; FC > 1.2, adjusted P < 0.05) were accompanied by clinical features in the case of CliProtRsf.

ProtRsf model was used to build prognostic survival curve of each patients resulting in Continuous risk ranking (Crank) and considered as recurrence risk. Crank scores were further used to stratify recurrence and the non-recurrence groups using threshold by averaging the mean Cranks of two groups. These threshold was further using the independent test cohort.

Reply:

Thank you for the summary.

Results

Line number 81- 109 (The overall study design.....during the postoperative follow-up) can be trimmed down to 1 paragraph and most of the data must be included in Study population and samples collection section

Reply:

According to the advice, we modified our text as follows.

Results section

“Clinical characteristics of our study population

The overall study design is demonstrated in Figure 1A. We enrolled 85 PPTC patients (PM) and 83 pediatric patients with benign nodules (P.B.) (Figure 1B), and their clinicopathological features were summarized in Supplementary Table 1. This cohort included 23 males and 62 females (male-to-female ratio of 1:2.7) with an average age of 15.6 ± 2.4 years and 15 males and 68 females (male-to-female ratio of 1:4.5) with an average age of 15.9 ± 1.9 years in PM and PB groups respectively. All patients were admitted to the hospital with a mass in the neck, and their average tumor size was 2.4 ± 1.3 cm in PM group, which is smaller than those in PB group 3.8 ± 1.3 cm. In PM group, the median follow-up time was 71 months (interquartile range 48-113), during which no death was reported. Lung metastasis was discovered in one patient before the operation, and no change was reported after the radioactive iodine (RAI) therapy. Postoperative structural recurrence occurred in 12 cases (average age of 14): ten ipsilateral cervical lymph node metastases and one contralateral cervical lymph node metastase. One case developed postoperative lung metastases. All the cases of lymph node metastases were reoperated. During the follow-up evaluations, we found that the lesions of the patients with lung metastases had shrunk after the RAI therapy, and no growth or mental retardation was detected in any patient. Finally, no hematological or other secondary solid primary tumors were found during the postoperative follow-up.”

Methods section

“Study population

In this retrospective study, we evaluated pediatric patients (≤ 18 years) with thyroid nodules, including 85 PM and 83 PB thyroid nodules, who underwent surgery in the First Hospital of China Medical University between November 2007 and April 2021. This study was approved by the Ethics Committee of the First Hospital of China Medical University with the study number 2021-287-2.

The exclusion criteria for PM were the following: (a) with a history of radiation exposure or

family history, (b) with poorly differentiated PTC, (c) loss of follow-up or incomplete clinical data, and (d) non-primary operation. We excluded uncertain malignant potential nodules for the PB group. We also included 66 AM patients with PTC to compare pediatric and adult thyroid cancer proteomic profiling. The detailed pediatric patient characteristics are listed in the **Supplementary Table 1**.

Preoperative pulmonary computed tomography (CT) showed that one patient had multiple metastases in the lung. All patients were surgically treated. Lobectomy and ipsilateral central lymph node dissections were performed in unilateral PTC. Total thyroidectomy was performed in patients with ETE, such as the invasion of nerves, blood vessels, or trachea. Patients with bilateral PTC underwent total thyroidectomy and bilateral central lymph node dissections. For PM patient group, 47 (55.29%) underwent lobectomy, and 38 (44.71%) had a total thyroidectomy. We recorded 16 cases (18.82%) with multifocal disease, 69 (81.18%) with lymph node metastases and 43 cases (43/85, 50.59%) of lateral cervical lymph node metastases in PM group. Postoperative treatment included thyroid-stimulating hormone inhibition and RAI therapies.

After the surgery, the patients were required to have follow-up visits every 3-6 months for the first year through cervical ultrasounds and thyroid functional examinations. The re-examination interval was then prolonged for patients with negative ultrasounds or CT, low serum thyroglobulin level, or no persistent disease. Disease remission and recurrence were determined according to the American Thyroid Association management guidelines^{11,12}. Disease remission was defined as two consecutive negative whole-body scans and ultrasounds. Due to inaccurate evaluation based on serum Tg and TgAb for patients with lobectomy, only structural recurrence was considered in this study. Disease recurrence was determined as a new disease in the thyroid bed or lymph nodes proven by cytology or histopathology, and/or confirmed by ultrasounds or CT scans, or distant metastases detected by whole-body scan. ”

Three clinical features are the risk factors of PPTC recurrence

Univariate Cox proportional hazard (CoxPH) model showed that age, TLNN, and LLNN may be risk factors for recurrence in pediatric patients. To determine the form of the age variable, the ten clinical features were next used as the inputs of multivariate CoxPH models.

Reply:

Thank you for your summary.

Protein qualification

Around 1272 proteins was removed with a missing value (NA) rate above 85%. CV of proteins were calculated cross the 16 pooled samples were mainly between 0.0 and 0.2, with a median of 0.0493 the CVs of the proteins across each pair of replicates were mostly less than 0.2, with medians of 0.0662, 0.0947, 0.1238, 0.0890, 0.0645 and 0.1123.

- Would recommend plotting correlation plot in which one half will be representing the intra sample correlation and other half inter sample correlation, thus one can determine the intra and inter sample correlation of individual sample.

Reply:

According to your suggestion, we plot the Pearson correlations for each paired sample. From the heatmap of correlation, we can find that the samples from the same group are well clustered, which indicates the substantial differences on proteotype. The low-risk PM samples are similar to the high-risk PMs based on the whole proteome.

Legend: Heatmap of Pearson correlation coefficients between each paired sample. Samples are labeled by the patient group. PM_Low and PM_high indicate the predicted risk of recurrence.

- One should mention the dimensionalities of data processing in terms of Normalization, Scaling and transformation

Reply:

In this study, we adopted a sequential strategy to preprocess our protein matrix. Firstly, to correct the plex-by-plex batch effects, we used internal reference scaling to convert the raw intensities into the relative abundances; then, after removing proteins with high missing proportions (> 85%), we imputed the missing values by *impseqrob* method, and finally, corrected the remaining batch effects by *ComBat* algorithm.

Proteomic differences among pediatric malignant, pediatric benign and adult malignant thyroid nodules

Around 243 was significant in (PM vs. PB) and 121 proteins were significant in (PM vs. AM) differentially expressed proteins (DEPs) with fold change (FC) > 1.5 and adjusted P < 0.05.

- Again the axis of Figure 3A, 3B and 3C are showing $-\log_{10}$ (Abundances of P value) vs $\log_2(\text{FC})$. Thus during the volcano plot (FC) or $\pm \log_2(\text{FC})$ was considered was not exclusive. One should mention this DEPs are both significant and differentially expressed as its fold change $(\text{FC}) \geq \pm 1.5$ and adjusted $P < 0.05$.

Reply:

Thanks for this suggestion. We have supplemented these DEPs derived from the FC 1.5 threshold in the caption of this figure.

"(C) The scatter diagram shows the FC distribution of the dysregulated proteins in two pairwise comparisons: PM vs. PB and PM vs. AM. Here, the DEP lists were derived from fold change threshold 1.5. "

- By lowering our FC threshold to 1.2, the number of DEPs increased to 1548 (PM vs. PB) and 1629 (PM vs. AM). What was the reason of lowering the FC, these are from FFPE tissue samples, and proteins must be significantly dysregulated as exceptionally observed for autoantibodies in serum/plasma samples.

Reply:

In this study, we used the tandem mass tag (TMT) labeling technique to quantify the proteome. The data we analyzed were adjusted by internal reference scaling firstly to mitigate the potential plex-to-plex batch effects, which means the protein abundances in our matrix were expression ratios rather than the intensities themselves. In each TMT 16-plex batch, there are 15 samples to be analyzed and one pooled sample. The relative protein abundance of 15 samples were represented by the ratios of the original intensities to the corresponding intensity of the pooled sample. The pooled samples are the same and act as a bridge between different batches. Therefore, compared to protein abundances measured using label-free quantification, these ratios have been compressed in terms of mean and scale. The fold changes derived from ratio expression are not as large as those from the original intensity, which is called Ratio Compression Effects¹. Thus, a lower threshold for fold change has been adopted in some published papers

using the TMT technique, such as \log_2 fold change 0.25 (fold change 1.189207)² and fold change 1.2³.

In the meantime, to avoid losing too many proteins by simple rules like FC, we adopted protein list from FC threshold 1.2 to give our model more freedom to decide by itself (though more time-consuming) which protein feature to use, which is a more precise way to build a more powerful model. In this way, the model would only select the most predictive protein combination. And the combinations derived from FC 1.2 threshold protein list would be much more and fully include those from FC 1.5 threshold, which leads to more (at least equally) powerful prediction model compared to that from FC 1.5 threshold.

- Authors have performed a differential proteome analysis between (PM vs. AM) and (PM vs. PB). To clarify the message for the scientific community, a functional clustering of differential expressed proteins between (PM vs. AM) and (PM vs. PB) is needed to be elaborated in main figure. This figure should complement the actual figure 3. Currently, this information is present in fragmented, volcano plots can be moved to figure S2A. The molecular functions their ORA or GSEA and their enrichment scores can be brought in Figure 3.

Reply:

Thank you for your comments. To reduce the fragmentation of information, we have placed all panels for the results of enrichment analysis and differential proteins (volcano plot) in the main Figure 3 in the updated version.

Immune infiltration and expression level of immune checkpoints in pediatric thyroid nodules

In-Silico tumour microenviron were studied using 'in-silico flow cytometry' CIBERSORTx. Seven types of immune cells were imputed namely CD8+ T cells ($P = 3.7 \times 10^{-12}$), macrophages ($P = 0.031$), dendritic cells ($P = 1.4 \times 10^{-5}$) and Treg cells ($P = 0.007$). CD8+ T cells and macrophages are increased in PM samples, while dendritic cells and Treg cells are reduced in PM

samples. CD8+ T cells and macrophages are increased in PM samples, while dendritic cells and Treg cells are reduced in PM samples which were validated using immunofluorescent staining.

Reply:

Thank you very much for the summary. We found that most regulated proteins in the comparison of PM vs PB are involved in the immune system. Therefore, we first performed CIBERSORT in-silico analysis to find the potential altered immune cells and then we validated the cells by immunofluorescent staining.

Analysis of 19 feature proteins

Random survival forest algorithm selected 19 proteins as features for the ProtRsf model, network analysis showed that 13 of the 19 protein features were directly or indirectly connected to disease.

- What about protein interactomes specifically modulated in each cohort comparison? Authors should complement their analysis including protein interactome networks, looking for specific hubs in each cohort comparison. To enrich the discussion section, are there specific transcription factors or upstream regulators that might be involved in the downstream modulation observed at proteome level in each cohort of PPTC comparison?

Reply:

To analyze the transcription factors or upstream regulators, we performed IPA with upstream module. There were four transcription regulators mapped with $P < 0.01$ for the 19-protein panel. The results are listed in the table below. We further searched the literature through PubMed and found that SREBF1 is a reported, prognostically relevant protein in thyroid cancer^{18,19}. The other proteins with $P < 0.01$, which have not been reported in studies related to thyroid cancer prognosis, are potential regulators that we identified. This part of the new results have been added to our revised manuscript.

Supplementary Table 7. Predicted transcription regulators of the 19 protein features

Molecule Type	Upstream Regulator	P value of overlap	Target Molecules in Dataset
transcription regulator	SREBF1	0.000977	GLDC, LGALS3, RNASET2
transcription regulator	ERCC6	0.00208	LGALS3
transcription regulator	CBFA2T2	0.00277	CHGA
transcription regulator	EOMES	0.00285	COL6A3, ITGA4

P values are calculated by Fisher's exact test.

- Authors can also perform targeted proteomic validation on selected protein features using SRM/PRM to see the trend of this proteins across each cohort of PPTC comparison.

Reply:

This is a good suggestion. In this case, we used TMT-labeled proteomics, the discovery proteomics strategy, which can get deep identifications to find the potentially altered pathways and functions. For further clinical translation, we are setting up a new multicenter prospective study to validate the protein panel by the cheaper and faster proteomic strategy, SRM. This part is ongoing and has not been included in the present study.

References

- 1 Savitski, M. M. *et al.* Measuring and managing ratio compression for accurate iTRAQ/TMT quantification. *J Proteome Res* **12**, 3586-3598, doi:10.1021/pr400098r (2013).
- 2 Shen, B. *et al.* Proteomic and Metabolomic Characterization of COVID-19 Patient Sera. *Cell* **182**, 59-72 e15, doi:10.1016/j.cell.2020.05.032 (2020).
- 3 Nie, X. *et al.* Multi-organ proteomic landscape of COVID-19 autopsies. *Cell* **184**, 775-791 e714, doi:10.1016/j.cell.2021.01.004 (2021).
- 4 Newman, A. M. *et al.* Determining cell type abundance and expression from bulk tissues with digital cytometry. *Nat Biotechnol* **37**, 773-782, doi:10.1038/s41587-019-0114-2 (2019).
- 5 Stewart, P. A. *et al.* Proteogenomic landscape of squamous cell lung cancer. *Nat Commun* **10**, 3578, doi:10.1038/s41467-019-11452-x (2019).
- 6 Wang, Y. *et al.* Longitudinal proteomic investigation of COVID-19 vaccination. *Protein Cell* **14**, 668-682, doi:10.1093/procel/pwad004 (2023).
- 7 Chen, Y. *et al.* Proteomic profiling of gastric cancer with peritoneal metastasis identifies a protein signature associated with immune microenvironment and patient outcome. *Gastric Cancer* **26**, 504-516, doi:10.1007/s10120-023-01379-0 (2023).
- 8 Newman, A. M. *et al.* Robust enumeration of cell subsets from tissue expression profiles. *Nat Methods* **12**, 453-457, doi:10.1038/nmeth.3337 (2015).
- 9 Iosef, C. *et al.* COVID-19 plasma proteome reveals novel temporal and cell-specific signatures for disease severity and high-precision disease management. *J Cell Mol Med* **27**, 141-157, doi:10.1111/jcmm.17622 (2023).
- 10 Iosef, C. *et al.* Plasma proteome of Long-COVID patients indicates HIF-mediated vasculo-proliferative disease with impact on brain and heart function. *J Transl Med* **21**, 377, doi:10.1186/s12967-023-04149-9 (2023).
- 11 Haugen, B. R. *et al.* 2015 American Thyroid Association Management Guidelines for Adult Patients with Thyroid Nodules and Differentiated Thyroid Cancer: The American Thyroid Association Guidelines Task Force on Thyroid Nodules and Differentiated Thyroid Cancer. *Thyroid* **26**, 1-133, doi:10.1089/thy.2015.0020 (2016).
- 12 Francis, G. L. *et al.* Management Guidelines for Children with Thyroid Nodules and Differentiated Thyroid Cancer. *Thyroid* **25**, 716-759, doi:10.1089/thy.2014.0460 (2015).
- 13 Memeh, K. *et al.* Total Thyroidectomy vs Thyroid Lobectomy for Localized Papillary Thyroid Cancer in Children: A Propensity-Matched Survival Analysis. *J Am Coll Surg* **233**, 39-49, doi:10.1016/j.jamcollsurg.2021.03.025 (2021).
- 14 Sudoko, C. K. *et al.* Thyroid Lobectomy for T1 Papillary Thyroid Carcinoma in Pediatric Patients. *JAMA Otolaryngol Head Neck Surg* **147**, 943-950, doi:10.1001/jamaoto.2021.2359 (2021).
- 15 Sugino, K. *et al.* Risk Stratification of Pediatric Patients with Differentiated Thyroid Cancer: Is Total Thyroidectomy Necessary for Patients at Any Risk? *Thyroid* **30**, 548-556, doi:10.1089/thy.2019.0231 (2020).
- 16 Baloch, Z. W. *et al.* Overview of the 2022 WHO Classification of Thyroid Neoplasms. *Endocr Pathol* **33**, 27-63, doi:10.1007/s12022-022-09707-3 (2022).
- 17 Jung, C. K., Bychkov, A. & Kakudo, K. Update from the 2022 World Health Organization

- Classification of Thyroid Tumors: A Standardized Diagnostic Approach. *Endocrinol Metab (Seoul)* **37**, 703-718, doi:10.3803/EnM.2022.1553 (2022).
- 18 Li, C. *et al.* SREBP1 as a potential biomarker predicts levothyroxine efficacy of differentiated thyroid cancer. *Biomed Pharmacother* **123**, 109791, doi:10.1016/j.biopha.2019.109791 (2020).
- 19 Kuo, C. Y. *et al.* SREBP1 promotes invasive phenotypes by upregulating CYR61/CTGF via the Hippo-YAP pathway. *Endocr Relat Cancer* **29**, 47-58, doi:10.1530/ERC-21-0256 (2021).